# Full-Atom Peptide Design
# with Geometric Latent Diffusion

**Xiangzhe Kong**[1,2]  **Yinjun Jia**[3]  **Wenbing Huang**[4*]  **Yang Liu**[1,2,5*]
[1]Dept. of Comp. Sci. & Tech., Tsinghua University
[2]Institute for AIR, Tsinghua University  [3]School of Life Sciences, Tsinghua University
[4]Gaoling School of Artificial Intelligence, Renmin University of China
[5]Shanghai Artificial Intelligence Laboratory, Shanghai, China

## Abstract

Peptide design plays a pivotal role in therapeutics, allowing brand new possibility to leverage target binding sites that are previously undruggable. Most existing methods are either inefficient or only concerned with the target-agnostic design of 1D sequences. In this paper, we propose a generative model for full-atom **Pep**tide design with **G**eometric **LA**tent **D**iffusion (PepGLAD) given the binding site. We first establish a benchmark consisting of both 1D sequences and 3D structures from Protein Data Bank (PDB) and literature for systematic evaluation. We then identify two major challenges of leveraging current diffusion-based models for peptide design: the full-atom geometry and the variable binding geometry. To tackle the first challenge, PepGLAD derives a variational autoencoder that first encodes full-atom residues of variable size into fixed-dimensional latent representations, and then decodes back to the residue space after conducting the diffusion process in the latent space. For the second issue, PepGLAD explores a receptor-specific affine transformation to convert the 3D coordinates into a shared standard space, enabling better generalization ability across different binding shapes. Experimental Results show that our method not only improves diversity and binding affinity significantly in the task of sequence-structure co-design, but also excels at recovering reference structures for binding conformation generation.

## 1 Introduction

Peptides are short chains of amino acids and acts as vital mediators of many protein-protein interactions in human cells. Designing functional peptides has attracted increasing attention in biological research and therapeutics, since the highly-flexible conformation space of peptides allows brand new possibility to target binding sites previously undruggable with antibodies or small molecules [17, 35]. The key of peptide design is to generate peptides that interact compactly with target proteins (see Figure 1), since they mostly exhibit flexible conformations [20] unless bound to these receptors [60].

Conventional simulation or searching algorithms rely on frequent calculations of physical energy functions [6, 9], which are inefficient and prone to poor local optimum. Recent advances illuminate the remarkable success of exploiting geometric deep generative models, particularly the equivariant diffusion models [45, 72], for molecule design [21, 39], antibody design [29, 44, 34] and protein design [64, 28], as well as latent diffusion models further enhancing the performance [71, 18]. Inspired by these successes, a natural idea is leveraging diffusion models for peptide design as well, which, yet, is challenging in two aspects. From the dataset aspect, existing databases (PepBDB [65], Propedia [46]) merely collect data from Protein Data Bank (PDB) [5], neither performing filters according to practical relevance [49] and redundancy, nor providing an adequate split for evaluation.

---

*Correspondence to Wenbing Huang <hwenbing@126.com>, Yang Liu <liuyang2011@tsinghua.edu.cn>

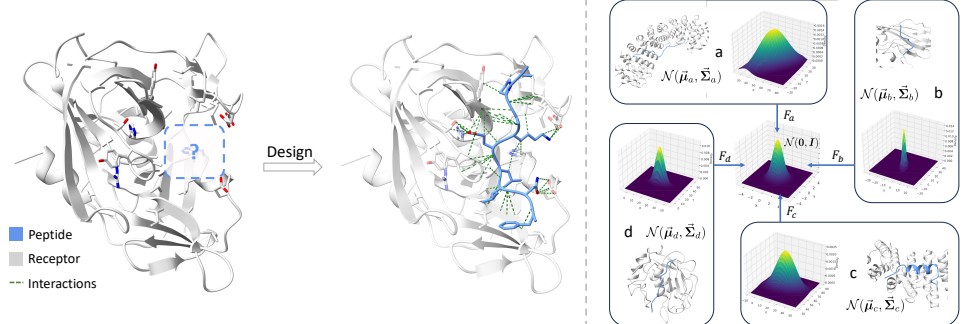

Figure 1: **Left**: Peptide design requires generating peptides that form compact interactions with the binding site on the receptor. The intricacy of protein-peptide interactions demands efficient exploration in the vast space for sequence-structure co-design. **Right**: Different binding sites (a, b, c, d) adopt disparate center offsets and geometric shapes, approximating variable 3D Gaussian distributions that deviate from $\mathcal{N}(\mathbf{0}, \boldsymbol{I})$. We propose to convert the geometry into a standard space approximating standard Gaussian, via an affine transformation derived from the binding site (§3.3).

Therefore, this paper first curates a benchmark from PDB [5] and the literature [59], and then systematically evaluates the generative models in terms of diversity, consistency, and binding affinity.

From the methodology aspect, it is nontrivial to adopt latent diffusion models to characterize the geometry of protein-peptide interactions. The first nontriviality stems from the *full-atom geometry*, which determines the comprehensive protein-peptide interactions in the atomic level, yet difficult to preserve. Throughout the generation process, the type of each amino acid always changes and thus requires us to generate different number of atoms, which is unfriendly to diffusion models that prefer fixed-size generation. Current latent diffusion models on molecules [71] or protein backbones [18] still tackle tasks with fixed number of atoms, thus leaving this challenge untouched. The second nontriviality lies in the *variable binding geometry*. Diffusion models are typically implemented directly in the data space, which might be suitable for regular data (*e.g.* images with fixed value range), yet ill-suited for our case on 3D coordinates where the value range is not fixed and even cursed with high variances due to the rich diversity in protein-peptide interactions. These variances define divergent target distributions of Gaussian with disparate expectation and covariance, which hinders the transferability of the diffusion process across different binding sites and thereby yields unsatisfactory generalization capability. Unfortunately, this point is seldom investigated previously.

To address the above problems, we propose a powerful model for full-atom **Pep**tide design with **G**eometric **LA**tent **D**iffusion (PepGLAD) with the following contributions:

- We construct a new benchmark from PDB and literature based on practical relevance and non-redundancy, then systematically evaluate available sequence-structure co-design models on the task of target-specific peptide design.

- To capture the *full-atom geometry*, we first learn a Variational AutoEncoder (VAE) to obtain a fixed-size latent representation (including a 3D coordinate and a hidden feature) for each residue of the input peptide, and then conduct the diffusion process in this latent space, both of which are conditioned on the binding site to better model protein-peptide interactions. Notably, the proposed design enables our model to accommodate full-atom input and output.

- Regarding the *variable binding geometry*, we derive a shared standard space from the binding sites by proposing a novel skill—receptor-specific affine transformation. Such affine transformation is computed by the center offset and covariance Cholesky decomposition of the binding site coordinates, serving as a mapping from the binding site distribution to standard Gaussian. With the affine transformation applied to both the binding sites and the peptides, we are able to project the shape of all complexes into approximately standard Gaussian distribution, which facilitates generalization to diverse binding sites.

Favorably, all the aforementioned models and processes meet the desired symmetry, *i.e.*, E(3)-equivariance, as proved by us. Experiments on sequence-structure co-design and complex conformation generation demonstrate the superiority of PepGLAD over the existing generative models.

## 2 Related Work

**Peptide Design** Conventional methods directly sample residues [6] or building blocks from libraries containing small fragments of proteins [26, 57, 9, 8], with guidance from delicate physical energy functions [2]. These methods are time-consuming and easy to be trapped by local optimum. Recent advances with deep generative models mainly focuses on target-agnostic 1D language models [48], antimicrobial peptides [13, 63], or a subtype of peptides with $\alpha$-helix [68, 69]. While geometric deep generative models are exhibiting notable potential in other domains of target-specific binder design (*e.g.* antibodies), their capability of target-specific peptide design remains unclear, which is the first problem we answer in this paper. Other contemporary work includes peptide design algorithms with flow matching frameworks [38, 40].

**Geometric Protein/Antibody Design** Protein design primarily aims to generate stable secondary or tertiary structures [27], where diffusion models demonstrate inspiring performance [66, 58, 3, 72]. In particular, RFDiffusion [64] first generates backbones via diffusion, and then designs the sequences through cycles of inverse folding and structure refining with empirical force fields. Chroma [28] adopts a similar strategy, but further explores controllable generative process with custom energy functions. Antibody design, encompassing a special family of proteins in the immune system to capture antigens, mainly focuses on inpainting complementarity-determining regions (CDRs) at the interface between the antigen and the framework [33, 34, 61], where the geometric diffusion models exhibit promising potential [44, 45] in co-designing sequence and structure. Unlike antibodies which are constrained by framework regions, peptides exhibit a more irregular binding pattern and greater flexibility, adapting to binding sites upon interaction [35]. Thus the target distributions are remarkably divergent on different binding sites, posing an urgent need for more robust generative modeling.

**Geometric Latent Diffusion Models** Diffusion models learn a denoising trajectory to generate desired data distribution from a prior distribution, commonly standard Gaussian [54, 55, 25]. Recent literature extends diffusion to 3D small molecules satisfying the E(3)-equivariance [70], which triggers subsequent advances in geometric design of macro molecules (*e.g.* antibody, protein) as aforementioned. Further efforts are made to latent diffusion models [53, 71, 18], which implement the generative process in the compressed latent space of pretrained auto-encoders, to improve the performance. Compared to the literature that either encodes atom-wise representation in the latent space for small molecule generation [71], or compress a fixed number of atoms into one latent node for protein backbone generation [18], we explore compression of the full-atom geometry by directly generating different residues with variable number of atoms in the latent space. Moreover, we propose a novel technique, namely the data-specific affine transformations, to enhance the generalization ability of diffusion models, which is barely explored before.

## 3 Our Method: PepGLAD

We first define the notations in the paper and formalize peptide design in §3.1. The overall workflow of our PepGLAD is presented in Figure 2, which consists of three modules: (1) An autoencoder that defines the joint latent space for sequences and structures conditioned on the full-atom context of the binding site (§3.2); (2) An affine transformation derived from the binding site to project the 3D geometry into a standard space approximating standard Gaussian distribution (§3.3); (3) A latent diffusion model trained on the standard latent space (§3.4). Finally, we summarize the training and the sampling procedures in §3.5.

### 3.1 Definitions and Notations

We represent binding sites and peptides as geometric graphs $\mathcal{G} = \{(x_i, \vec{X}_i)\}$, where each node $i$ is a residue with its amino acid type $x_i$ and the coordinates of all its $c_i$ atoms $\vec{X}_i \in \mathbb{R}^{c_i \times 3}$. In later sections, we use the simplified notations $i \in \mathcal{G}$ to denote that a node $i$ is in the geometric graph $\mathcal{G}$, and $|\mathcal{G}|$ to denote the total number of nodes in $\mathcal{G}$. We use $\mathcal{G}_p$ and $\mathcal{G}_b$ to represent the geometric graph of the peptide and the binding site, respectively. In this work, the binding site incorporates residues on the target protein within 10Å distances to the peptide residues based on $C_\beta$ atoms which alleviates leakage of the side-chain interactions. Note that the threshold (10Å) is chosen to be large to better reduce leakage of the peptide geometry.

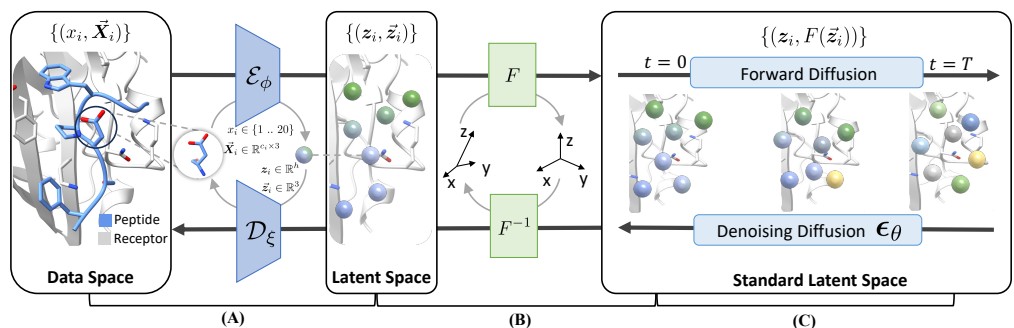

Figure 2: Overall architecture of PepGLAD. **(A)** Variational AutoEncoder (§3.2): compressing the sequence and the structure $\{(x_i, \vec{\boldsymbol{X}}_i)\}$ of the peptide into the latent space $\{(\boldsymbol{z}_i, \vec{\boldsymbol{z}}_i)\}$ with the encoder $\mathcal{E}_\phi$, and decoding the sequence and full-atom geometry from the latent states with the decoder $\mathcal{D}_\xi$. **(B)** Affine Transformation $F$ (§3.3): projecting the geometry to approximately $\mathcal{N}(\boldsymbol{0}, \boldsymbol{I})$ via the receptor-specific affine transformation derived from the binding site, and recovering the data geometry with the inverse of $F$ after the diffusion generative process. **(C)** Latent Diffusion (§3.4): jointly generating $\boldsymbol{z}_i$ and $\vec{\boldsymbol{z}}_i$ in the standard latent space.

**Task Definition** Given the binding site $\mathcal{G}_b$, we aim to obtain a generative model $p_\theta$ conforming to the distribution of binding peptides $q(\mathcal{G}_p|\mathcal{G}_b)$.

## 3.2 Variational AutoEncoder

The autoencoder [62] consists of an encoder $\mathcal{E}_\phi$ that encodes the peptide $\mathcal{G}_p$ in the presence of the binding site $\mathcal{G}_b$ into a latent state $\mathcal{G}_z$, and a decoder $\mathcal{D}_\xi$ that reconstructs the peptide from the latent state to obtain $\mathcal{G}'_p = \{(x'_i, \vec{\boldsymbol{X}}'_i)\}$. To encourage $\mathcal{E}_\phi$ to learn contextual representations of residues, we corrupt 25% of the residues in $\mathcal{G}_p$ with a [MASK] type to obtain $\tilde{\mathcal{G}}_p$ as the input:

$$\mathcal{G}_z = \mathcal{E}_\phi(\tilde{\mathcal{G}}_p, \mathcal{G}_b), \qquad \mathcal{G}'_p = \mathcal{D}_\xi(\mathcal{G}_z, \mathcal{G}_b), \tag{1}$$

where $\mathcal{G}_z = \{(\boldsymbol{z}_i, \vec{\boldsymbol{z}}_i)|i \in \mathcal{G}_p\}$ contains the latent states $\boldsymbol{z}_i \in \mathbb{R}^h$ ($h = 8$ in this paper) and $\vec{\boldsymbol{z}}_i \in \mathbb{R}^3$ sampled from the encoded distribution $\mathcal{N}(\boldsymbol{z}_i; \boldsymbol{\mu}_i, \boldsymbol{\sigma}_i)$ and $\mathcal{N}(\vec{\boldsymbol{z}}_i; \vec{\boldsymbol{\mu}}_i, \vec{\boldsymbol{\sigma}}_i)$ using the reparameterization trick [32]. We borrow the adaptive multi-channel equivariant encoder in dyMEAN [34] for both $\mathcal{E}_\phi$ and $\mathcal{D}_\xi$ to capture the full-atom geometry. In the decoder $\mathcal{D}_\xi$, we factorize the joint distribution of sequences and structures as follows:

$$p_\xi(x'_i, \vec{\boldsymbol{X}}'_i|\mathcal{G}_z, \mathcal{G}_b) = p_{\xi_1}(x'_i|\mathcal{G}_z, \mathcal{G}_b)p_{\xi_2}(\vec{\boldsymbol{X}}'_i|x'_i, \mathcal{G}_z, \mathcal{G}_b), \tag{2}$$

where the sequence is first decoded and then the all-atom geometry, initialized with replications of $\vec{\boldsymbol{z}}_i$, is reconstructed. The training objective of the autoencoder consists of the reconstruction loss $\mathcal{L}_{recon}$ and the KL divergence $\mathcal{L}_{KL}$ to constrain the latent space. The reconstruction loss includes cross entropy on the residue types, mean square error (MSE) on the full-atom structures, and an auxiliary loss $\mathcal{L}_{aux}$ on bond lengths and angles [31]:

$$\mathcal{L}_{recon}(i) = H(p(x_i), p(x'_i)) + \text{MSE}(\vec{\boldsymbol{X}}_i, \vec{\boldsymbol{X}}'_i) + \mathcal{L}_{aux}(i), \tag{3}$$

where $H$ denotes cross entropy. We include details of $\mathcal{L}_{aux}$ in Appendix A. The KL divergence constrains $\boldsymbol{z}_i$ and $\vec{\boldsymbol{z}}_i$ with the prior $\mathcal{N}(\boldsymbol{0}, \boldsymbol{I})$ and $\mathcal{N}(\vec{\boldsymbol{r}}_i, \boldsymbol{I})$, respectively, where $\vec{\boldsymbol{r}}_i$ denotes the coordinate of the alpha carbon ($C_\alpha$) in node $i$:

$$\mathcal{L}_{KL}(i) = \lambda_1 \cdot D_{\text{KL}}(\mathcal{N}(\boldsymbol{0}, \boldsymbol{I})\|\mathcal{N}(\boldsymbol{\mu}_i, \text{diag}(\boldsymbol{\sigma}_i))) + \lambda_2 \cdot D_{\text{KL}}(\mathcal{N}(\vec{\boldsymbol{r}}_i, \boldsymbol{I})\|\mathcal{N}(\vec{\boldsymbol{\mu}}_i, \text{diag}(\vec{\boldsymbol{\sigma}}_i))), \tag{4}$$

where $D_{\text{KL}}$ denotes the KL divergence, $\lambda_1$ and $\lambda_2$ reweight the contraints on the sequence and the structure, respectively. $\mathcal{L}_{KL}$ prevents the scale of $\boldsymbol{z}_i$ from exploding and constrains $\vec{\boldsymbol{z}}_i$ around $C_\alpha$ to retain necessary geometric information. Such regularization also helps ensure consistent scales between the peptide latent coordinates and the pocket, mitigating potential issues arising from their different levels of abstraction. Then we have the overall training objective of the variational autoencoder as follows:

$$\mathcal{L}_{AE} = \sum\nolimits_{i \in \mathcal{G}_p}(\mathcal{L}_{recon}(i) + \mathcal{L}_{KL}(i))/|\mathcal{G}_p|. \tag{5}$$

We have explored E(3)-invariant latent space, which appears to have difficulties in reconstructing the full-atom structures since it lacks information of geometric interactions with the pocket atoms (Appendix F).

## 3.3 Receptor-Specific Affine Transformation

With the latent space given by the autoencoder, we further exploit a standard space obtained from receptor-specific affine transformations, which enhances the transferability of diffusions on disparate binding sites (see Figure 1). Most peptides fold into complementary shape upon binding on the receptor [60, 41]. Thus, the target distribution is inherently characterized by the shape of the binding site. Given the wide disparity in binding geometries, directly implementing diffusion in the data space yields minimal transferability among different binding sites. To address this deficiency, we propose to implement the diffusion process on a shared standard space converted via an affine transformation derived from the binding site. Formally, denoting the $C_\alpha$ coordinates of the residues in a given binding site $\mathcal{G}_b$ as $\vec{R} \in \mathbb{R}^{3 \times |\mathcal{G}_b|}$, we can derive their center $\vec{\mu} = \mathbb{E}[\vec{R}] \in \mathbb{R}^3$ and covariance $\vec{\Sigma} = \text{Cov}(\vec{R}, \vec{R}) \in \mathbb{R}^{3 \times 3}$, so that these coordinates can be regarded as sampled from the distribution $\mathcal{N}(\vec{\mu}, \vec{\Sigma})$. We then calculate the Cholesky decomposition [19] of $\vec{\Sigma}$:

$$\vec{\Sigma} = \vec{L}\vec{L}^\top, \vec{L} \in \mathbb{R}^{3 \times 3}, \tag{6}$$

where $\vec{L}$ is a lower triangular matrix. $\vec{L}$ is unique [19] and invertible since the covariance matrix is a real-valued symmetric positive-definite matrix[2]. Then we can define the affine transformation $F : \mathbb{R}^3 \to \mathbb{R}^3$, which enables the projection of the geometry into the standard space approximating standard Gaussian $F(\vec{R}) \sim \mathcal{N}(\mathbf{0}, \mathbf{I})$. Further, we can easily obtain the inverse of $F$ as:

$$F(\vec{x}) = \vec{L}^{-1}(\vec{x} - \vec{\mu}), \qquad F^{-1}(\vec{x}) = \vec{L}\vec{x} + \vec{\mu}. \tag{7}$$

With the above definitions, for each given binding site $\mathcal{G}_b$, we transform the geometry via the derived $F$ to obtain the standard space, where the diffusion model is implemented, and recover the original geometry with $F^{-1}$ (see Figure 2) after generation. Notably, we have the following proposition to ensure that the equivariance is maintained under the proposed affine transformation with scalarization-based equivariant GNNs [23, 16]:

**Proposition 3.1.** *Denote the invariant and equivariant outputs from a scalarization-based E(3)-equivariant GNN as $f(\{h_i, \vec{x}_i\})$ and $\vec{f}(\{h_i, \vec{x}_i\})$, respectively. With the definition of $F$ in Eq. 7, $\forall g \in E(3)$, we have $f(\{h_i, F(\vec{x}_i)\}) = f(\{h_i, F_g(g \cdot \vec{x}_i)\})$ and $g \cdot F^{-1}(\vec{f}(\{h_i, F(\vec{x}_i)\})) = F_g^{-1}(\vec{f}(\{h_i, F_g(g \cdot \vec{x}_i)\}))$, where $F_g$ is derived on the coordinates transformed by $g$. Namely, the E(3)-equivariance is preserved if we implement the GNN on the standard space and recover the original geometry from the outputs.*

The proof is in Appendix B. This is vital since it indicates the Markov kernel is E(3)-equivariant, and thus ensures the E(3)-invariance of the probability density in the diffusion process [70]. Note that our variational autoencoder (§ 3.2) and latent diffusion model (§ 3.4) are already designed to be equivariant even without the proposed affine transformation here. The purpose of defining such component is to encourage better generalization of the diffusion processes. Indeed, it is nontrivial to analyze whether such an implementation will break the equivariance of our workflow. Luckily, Proposition 3.1 manages to prove that scalarization-based equivariant networks [23], which is used in our autoencoder and diffusion model, are seamlessly compatible with such affine transformation, naturally preserving equivariance without any requirements of adaption.

## 3.4 Geometric Latent Diffusion Model

With the aforementioned preparations, the discrete residue types are encoded as continuous latent representations $\{z_i\}$, and the full-atom geometry is also compressed and standardized into 3D vectors $\{\vec{z}_i\} \sim \mathcal{N}(\mathbf{0}, \mathbf{I})$. Therefore, we are ready to implement a diffusion model on the standard latent space to generate $z_i$ and $\vec{z}_i$. The forward diffusion process gradually adds noise to the data from $t = 0$ to $t = T$, resulting in the prior distribution $\mathcal{N}(\mathbf{0}, \mathbf{I})$. The reverse diffusion process

---

[2]The binding site has at least 3 nonoverlapping nodes, namely $\text{rank}(\vec{R}) = 3$, thus we can ignore the corner case of semi-positive definite matrices.

generates data distribution by iteratively denosing the distribution from $t = T$ to $t = 0$. We denote $\vec{u}_i^t = [\mathbf{z}_i^t, \vec{z}_i^t]$ and $\mathcal{G}_z^t = \{(\mathbf{z}_i^t, \vec{z}_i^t)\}$ as the intermediate state for node $i$ and the entire peptide at time step $t$, respectively. For simplicity, we assume both $\mathcal{G}_z^t$ and the binding site $\mathcal{G}_b$ are already standardized via the transformation $F_b$ in Eq. 7. Then we have the forward process as:

$$q(\vec{u}_i^t | \vec{u}_i^{t-1}) = \mathcal{N}(\vec{u}_i^t; \sqrt{1 - \beta^t} \cdot \vec{u}_i^{t-1}, \beta^t \mathbf{I}), \tag{8}$$

$$q(\vec{u}_i^t | \vec{u}_i^0) = \mathcal{N}(\vec{u}_i^t; \sqrt{\bar{\alpha}^t} \cdot \vec{u}_i^0, (1 - \bar{\alpha}^t)\mathbf{I}), \tag{9}$$

where $\beta^t$ is the noise scale increasing with the timestep from 0 to 1 conforming to the cosine schedule [51], and $\bar{\alpha}^t = \prod_{s=1}^{s=t}(1 - \beta^s)$. Then the state at timestep $t$ can be sampled as:

$$\vec{u}_i^t = \sqrt{\bar{\alpha}^t}\vec{u}_i^0 + (1 - \bar{\alpha}^t)\boldsymbol{\epsilon}_i, \tag{10}$$

where $\boldsymbol{\epsilon}_i \sim \mathcal{N}(\mathbf{0}, \mathbf{I})$. Following Ho et al. [25], the reverse process can be defined with the reparameterization trick as:

$$p_\theta(\vec{u}_i^{t-1} | \mathcal{G}_z^t, \mathcal{G}_b) = \mathcal{N}(\vec{u}_i^{t-1}; \vec{\boldsymbol{\mu}}_\theta(\mathcal{G}_z^t, \mathcal{G}_b), \beta^t \mathbf{I}), \tag{11}$$

$$\vec{\boldsymbol{\mu}}_\theta(\mathcal{G}_z^t, \mathcal{G}_b) = \frac{1}{\sqrt{\alpha^t}}(\vec{u}_i^t - \frac{\beta^t}{\sqrt{1 - \bar{\alpha}^t}}\boldsymbol{\epsilon}_\theta(\mathcal{G}_z^t, \mathcal{G}_b, t)[i]), \tag{12}$$

where $\alpha^t = 1 - \beta^t$, and $\boldsymbol{\epsilon}_\theta$ is the denoising network also implemented with the equivariant adaptive multi-channel equivariant encoder in dyMEAN [34] to retain full-atom context of the binding site during generation and preserve the equivariance under affine transformations (Proposition 3.1). Finally, we have the objective at time step $t$ as MSE between the predicted noise and the added noise in Eq. 10, as well as the overall training objective $\mathcal{L}_{LDM}$ as the expectation with respect to $t$:

$$\mathcal{L}_{LDM} = \mathbb{E}_{t \sim \text{Uniform}(1...T)}[\sum_i \|\boldsymbol{\epsilon}_i - \boldsymbol{\epsilon}_\theta(\mathcal{G}_z^t, \mathcal{G}_b, t)[i]\|^2 / |\mathcal{G}_z^t|]. \tag{13}$$

### 3.5 Training and Sampling

**Training** The training of our PepGLAD can be divided into two phases where a variational autoencoder is first trained and then a diffusion model is trained on the standard latent space. We provide the overall training procedure in Algorithm 1 (see Appendix D). Note that a smooth and informative latent space is necessary for the consecutive training of the diffusion model, thus we resort to unsupervised data from protein fragments apart from the limited protein-peptide complexes for training the autoencoder, which we describe in Appendix E.

**Sampling in Ordered Subspace** The sampling procedure includes generative diffusion process on the standard latent states, recovering the original geometry with the inverse of $F$ in Eq. 7, and decoding the sequence as well as the full-atom structure of the peptide (see Algorithm 2 in Appendix D). A problem here is that the unordered nature of graphs is not compatible with the sequential nature of peptides, thus the generated residues may have arbitrary permutation on the sequence order. Inspired by the concept of classifier-guided sampling [15], we first assign an arbitrary permutation $\mathcal{P}$ on the sequence order to the nodes. Then we steer the sampling procedure towards the desired subspace conforming to $\mathcal{P}$ with the following empirical classifier $p(1|\{\vec{z}_i^t\})$, which estimates the probability of the current coordinates belonging to the desired subspace:

$$p(1|\{\vec{z}_i^t\}) = \exp(-\sum_{\mathcal{P}(i) - \mathcal{P}(j) = 1} E(\|\vec{z}_i^t - \vec{z}_j^t\|)), \tag{14}$$

$$E(d) = \begin{cases} d - (\mu_d + 3\sigma_d), & d > \mu_d + 3\sigma_d, \\ (\mu_d - 3\sigma_d) - d, & d < \mu_d - 3\sigma_d, \\ 0, & \text{otherwise}, \end{cases} \tag{15}$$

where $\mu_d$ and $\sigma_d$ are the mean and variance of the distances of adjacent residues in the latent space measured from the training set. Intuitively, this classifier gives higher confidence if the adjacent (defined by $\mathcal{P}$) residues are within reasonable distances aligning with the statistics from the training set. Nevertheless, the effect of the guidance is relatively minor, which is only a technical trick to enhance the robustness. We provide more details in Appendix G.

## 4 Experiments

### 4.1 Setup

**Task** We evaluate our PepGLAD and baselines on the following tasks: (1) **sequence-structure co-design** (§4.2) aims to generate both the sequence and the structure of the peptide given the specific

binding site on the receptor (*i.e.* protein). (2) **Binding Conformation Generation** (§4.3) requires to generate the binding state of the peptide given its sequence and the binding site of interest.

**Dataset** We first extract all dimers from the Protein Data Bank (PDB) [5] and select the complexes with a receptor longer than 30 residues and a ligand between 4 to 25 residues [59]. Then we remove the duplicated complexes with the criterion that both the receptor and the peptide has a sequence identity over 90% [56], after which 6105 non-redundant complexes are obtained. To achieve the cross-target generalization test, we utilize the large non-redundant dataset (LNR) from Tsaban et al. [59] as the test set, which contains 93 protein-peptide complexes with canonical amino acids curated by domain experts. We then cluster the data by receptor with a sequence identity threshold of over 40%, and remove the complexes sharing the same clusters with those from the test set. Finally, the remaining data are randomly split based on clustering results into training and validation sets, yielding a new bechmark calling **PepBench**. Further, we exploit 70k unsupervised data from protein fragments (**ProtFrag**) to facilitate training of the variational autoencoder. We also implement a split on **PepBDB** [65] based on clustering results for evaluation. We show details and statistics of these datasets in Appendix E.

**Baselines** We first borrow three baselines from the antibody design domain. **HSRN** [30] autoregressively decodes the sequence while keeps refining the structure hierarchically, from the $C_\alpha$ to other atoms. **dyMEAN** [34] is equipped with an full-atom geometric encoder and exploits iterative non-autoregressive generation. **DiffAb** [44] jointly diffuses on the categorical residue type, the coordinate of $C_\alpha$ as well as the orientation of each residue. Next, we explore two baselines from the general protein design. **RFDiffusion** [64] exploits a pipeline that first generates the backbone via diffusion and then alternates between inverse folding [14] and structure refining based on a physical energy function [2]. **AlphaFold 2** [31] is the well-known model for protein folding, which also shows certain abilities on peptide conformation prediction [59]. We also include two traditional methods. **AnchorExtension** [26] designs peptides by first docking an existing scaffold to the binding site, and then optimizing the peptide with cycles of mutations guided by energy functions. **FlexPepDock** [42] is designed for flexible peptide docking via optimization in the landscape of a physical energy function [2]. Implementation details are provided in Appendix I.

## 4.2 Sequence-Structure Co-Design

**Metrics** A favorable generative model should produce diverse candidates while maintaining fidelity to the desired distribution. To comprehensively evaluate the models, we generate 40 candidates for each receptor and employ the following metrics: (1) **Diversity**. Inspired by [72], we measure the diversity via unique clusters of sequences and structures. Specifically, we hierarchically cluster the structures based on pair-wise root mean square deviation (RMSD) of $C_\alpha$. The diversity of structures $\text{Div}_{struct}$ is defined as the number of clusters versus the number of candidates. A similar procedure can be applied to the sequences to obtain $\text{Div}_{seq}$, utilizing the similarity [36] derived from alignment [24]. Then the co-design diversity is $\sqrt{\text{Div}_{seq}\text{Div}_{struct}}$. (2) **Consistency**. We measure how well the models learn the 1D&3D joint distribution by the sequence-structure consistency, quantified via Cramér's V [12] association between the clustering labels (as in Diversity) of the sequences and the structures. High consistency indicates that candidates with similar sequences also have similar structures, implying that the generative model effectively captures the dependency between 1D and 3D. (3) **$\Delta G$**. Aligned with the literature [34, 44], we employ the binding energy (kcal/mol) provided by Rosetta [2], a widely-used suite for biomolecular modeling with physical energy functions, to evaluate the binding affinity of the generated candidates. Lower $\Delta G$ indicates stronger binding between the peptide and the target. (4) **Success**. We report the proportion of successful designs (*i.e.* $\Delta G < 0$, indicating no severe atomic clashes or twisted conformations) among all the candidates.

Table 1: Evaluation on sequence-structure co-design. On each target, 40 candidates are generated for evaluation. Div. and Con. are abbreviations for diversity and consistency, respectively.

| Model | PepBench | | | | PepBDB | | | |
|---|---|---|---|---|---|---|---|---|
| | Div.($\uparrow$) | Con.($\uparrow$) | $\Delta G$($\downarrow$) | Success | Div.($\uparrow$) | Con.($\uparrow$) | $\Delta G$($\downarrow$) | Success |
| Test Set | - | - | -35.25 | 95.70% | - | - | -35.96 | 95.79% |
| HSRN[3] | 0.158 | 0.0 | $\geq 0$ | 10.46% | 0.111 | 0.0 | $\geq 0$ | 10.86% |
| dyMEAN | 0.150 | 0.0 | -2.26 | 14.60% | 0.150 | 0.0 | -1.92 | 6.26% |
| DiffAb | 0.427 | 0.670 | -21.20 | 49.87% | 0.269 | 0.463 | -18.40 | 41.45% |
| PepGLAD (ours) | **0.506** | **0.789** | **-21.94** | **55.97%** | **0.692** | **0.923** | **-21.53** | **48.47%** |

For all metrics except $\Delta G$, we first compute values for each receptor individually and then average the results across different receptors. For $\Delta G$, we identify the best candidate on each receptor as the outputs and report the median value across different receptors. Details about the metrics are provided in Appendix H, including discussion on AAR (Appendix H.1) and consistency (Appendix H.2).

**Results** Table 1 illustrates that our PepGLAD generates significantly more diversified and consistent peptides with better binding energy and success rates compared to the baselines. When benchmarking HSRN, dyMEAN, and DiffAb, which perform well on antibody CDR design, we observe a notable performance gap between non-diffusion baselines (*i.e.* HSRN, dyMEAN) and the diffusion-based baseline (*i.e.* DiffAb), suggesting the higher complexity in peptide design and the need for stronger modeling capabilities. Compared to DiffAb, which operates on categorical residue types, $C_\alpha$ coordinates and orientations, our PepGLAD (1) better captures the dependency between sequence and structure, as indicated by higher diversity and consistency, since diffusion is implemented on the latent space where the representation of sequence and structure are nicely correlated by the autoencoder; (2) more effectively captures the intricate protein-peptide interactions, demonstrated by better $\Delta G$ and success rates, since we leverage the full-atom context of the binding site and enhances generalization capability by converting the geometry into a standard space. We showcase two candidates designed by our PepGLAD with favorable binding energy given by Rosetta in Figure 3. Furthermore, the diversity within successful designs is 0.632, which is higher than that of all designs (0.506), indicating the high structural flexibility of peptides upon successful binding.

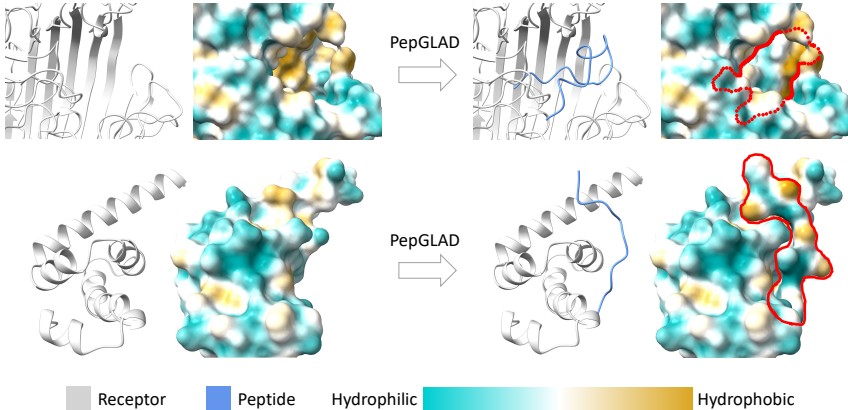

Figure 3: **Top**: A generated candidate confined within the binding site (PDB=4cu4, $\Delta G$=-34.21). **Bottom**: A generated candidate with complementary shape to the binding site (PDB=3pkn, $\Delta G$=-33.32). Both candidates form compact interactions at the interface.

We also evaluate our PepGLAD against two sophisticated pipeline systems in Table 2. The traditional method (*i.e.* AnchorExtension) is limited by low efficiency, thus we can only afford outputting 10 candidates for each receptor. For a relatively fair comparison with RFDiffusion, we refine the structure of the generated candidates using the empirical force field in RFDiffusion. However, the comparison may still disadvantage our PepGLAD, given that RFDiffusion is finetuned from a model pretrained on a large-scale dataset [64]. Nevertheless, as demonstrated in Table 2, our model still exhibits marvelous superiority on diversity, consistency, and success rate, while achieving competitive binding energy $\Delta G$, with obviously higher efficiency.

Table 2: Evaluation on sequence-structure co-design with two well-established systems. Time cost is measured as the total time spent divided by the number of designed candites.

| Model | Div.($\uparrow$) | Con.($\uparrow$) | $\Delta G$($\downarrow$) | Success | Time |
|---|---|---|---|---|---|
| AnchorExtension | 0.245 | 0.423 | -26.80 | 84.30% | 735s |
| RFDiffusion | 0.259 | 0.696 | **-33.82** | 79.68% | 61s |
| PepGLAD (ours) | **0.506** | **0.789** | -29.36 | **92.82%** | **3s** |

---

[3]HSRN and dyMEAN generate homogeneous structures that are clustered together yet still sample very different sequences, leading to zero association between squence and structure.

### 4.3 Binding Conformation Generation

**Metrics**  For each receptor, we generate 10 candidates and report the median value of the following metrics across different receptors to measure how well the generated distribution can recover the reference conformation: (1) $\mathbf{RMSD}_{C_\alpha}$: Root mean square deviation on the coordinates of $C_\alpha$ between a candidate and a reference structure with the unit Å. (2) $\mathbf{RMSD}_{atom}$: RMSD on all atoms to measure the quality of the full-atom geometry. (3) **DockQ** [4]. A comprehensive metric evaluating the full-atom similarity on the interface between a candidate and a reference complex. It ranges from 0 to 1, with values above 0.23 and 0.49 considered as acceptable and medium quality, respectively.

Table 3: Evaluation on binding conformation generation. On each target, 10 candidates are generated to calculate the optimal recall of the reference conformation.

| Model | PepBench | | | PepBDB | | |
|---|---|---|---|---|---|---|
| | $\text{RMSD}_{C_\alpha}(\downarrow)$ | $\text{RMSD}_{atom}(\downarrow)$ | DockQ($\uparrow$) | $\text{RMSD}_{C_\alpha}(\downarrow)$ | $\text{RMSD}_{atom}(\downarrow)$ | DockQ($\uparrow$) |
| FlexPepDock | 6.43 | 7.52 | 0.393 | - | - | - |
| AlphaFold 2 | 8.49 | 9.20 | 0.355 | - | - | - |
| dyMEAN | 7.96 | 8.35 | 0.374 | 17.64 | 17.56 | 0.142 |
| HSRN | 6.02 | 7.59 | 0.508 | 9.28 | 9.72 | 0.394 |
| DiffAb | 4.23 | 7.60 | 0.586 | 13.96 | 13.12 | 0.236 |
| PepGLAD (ours) | **4.09** | **5.30** | **0.592** | **8.87** | **8.62** | **0.403** |

**Results**  As shown in Table 3, our PepGLAD surpasses all the baselines in terms of both $\text{RMSD}_{C_\alpha}$ and DockQ by a large margin, highlighting the superiority of incorporating the full-atom context and the binding-site shape into the latent diffusion process. Additionally, we present the distribution of the best $\text{RMSD}_{C_\alpha}$ on different test receptors using box plots and showcase a generated conformation highly resembling the reference in Figure 4. The distribution reveals that our model achieves favorable performance on $\text{RMSD}_{C_\alpha}$ with lower variance on the test set compared to other baselines, exhibiting robust generalization ability across disparate binding sites.

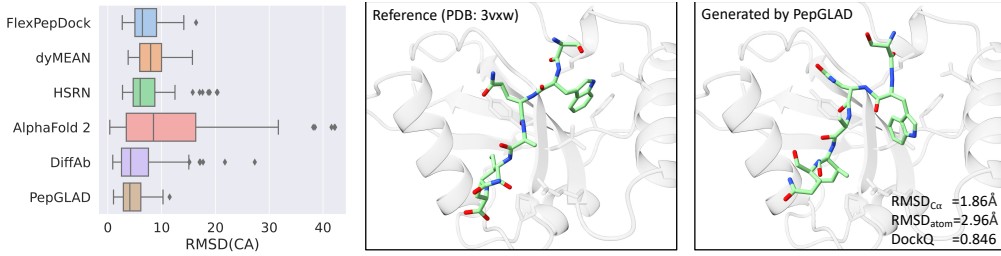

Figure 4: The distribution of $\text{RMSD}_{C_\alpha}$ on the test set of PepBench and a visualized sample.

## 5 Analysis

We conduct the following ablations: the full-atom geometry (**Full-Atom**); the affine transformation (**Affine**); the unsupervised data from protein fragments (**ProtFrag**) and the mask policy (**Mask**) when training the autoencoder. Note that generative performance is assessed from various aspects, and improvement in one aspect at the disproportionate expense of others might be meaningless. Thus, we additionally compute the average of all the metrics to evaluate the comprehensive effect of each module, where $\Delta G$ is normalized by the statistics on the test set. Table 4 demonstrates the following observations: (1) Discarding the full-atom context results in a significant degradation on all metrics, especially the success rate, implying the necessity of the full-atom context in capturing the intricate protein-peptide interactions; (2) Implementing the diffusion directly on the data space without the proposed affine transformation incurs a notably adverse impact on all metrics, indicating the remarkable enhancement on the generalization capability made by the affine transformation; (3) Training without the unsupervised data leads to a less informative latent space, exerting a negative effect on the binding energy and success rate; (4) Removal of the mask policy reduces the correlation between sequence and structure in the latent space, thus harms the consistency.

Table 4: Ablations on different components. Avg. computes the average of all metrics, where $\Delta G$ is first normalized by the median value of the references on test set.

| Ablations | Div.($\uparrow$) | Con.($\uparrow$) | $\Delta G(\downarrow)$ | Success | Avg. |
|---|---|---|---|---|---|
| PepGLAD | 0.506 | 0.789 | -21.94 | 55.97% | **0.619** |
| w/o Full-Atom | 0.441 | 0.751 | -20.87 | 51.18% | 0.574 |
| w/o Affine | 0.450 | 0.740 | -19.08 | 52.39% | 0.564 |
| w/o ProtFrag | 0.535 | 0.760 | -20.16 | 52.15% | 0.597 |
| w/o Mask | 0.422 | 0.741 | -20.45 | 57.44% | 0.579 |

## 6   Limitations

Despite the promising results, we acknowledge several limitations which might be addressed by future work. First, the binding affinity assessment relies on the Rosetta scoring function as a proxy for wetlab experiments. There may be discrepancies between the predicted and actual binding energies. The ultimate test of a peptide utility is its performance *in vivo*, which is too costly for large-scale evaluation. Nevertheless, this is a problem confronting the entire community, and we hope future research might propose more reliable *in silico* proxies to bridge the gap. Second, while this paper addresses peptide design from the aspect of proteins, it might also be reasonable to think from the aspect of small molecules if the peptides are short enough. Under such circumstances, it is also beneficial to further explore counterparts of methods for small molecule design [43, 52, 39], which we leave for future work.

## 7   Conclusion

In this paper, we first assemble a dataset from Protein Data Bank (PDB) and literature to benchmark generative models on target-specific peptide design in terms of diversity, consistency, and binding energy. Subsequently, we propose PepGLAD, a powerful diffusion-based model for full-atom peptide design. In particular, we explore diffusion on the latent space where the sequence and the full-atom structure are jointly encoded by a variational autoencoder. We further propose a receptor-specific affine transformation technique to project variable geometries in the data space into a standard space, which enhances the transferability of diffusion processes on disparate binding sites. Our PepGLAD outperforms the existing models on sequence-structure co-design and binding conformation generation, exhibiting high generalization across diverse binding sites. Our work represents a pioneering effort in the exploration of deep generative models for simultaneous design of 1D sequences and 3D structures of peptides, which could inspire future research in this field.

## Software and Data

The curated PepBench and ProtFrag are available at `https://zenodo.org/records/13373108`. The codes for our PepGLAD are open-sourced at `https://github.com/THUNLP-MT/PepGLAD`.

## Impact Statements

This paper aims to advance the field of peptide design through the construction of a benchmark and the development of a novel latent diffusion model, PepGLAD, which addresses key limitations in current methods. Our work represents a step forward in computational peptide design, with the potential to impact both scientific research and practical applications in various domains. For instance, more precise peptide design could lead to enhanced drugs in the pharmaceutical industry, and could facilitate the creation of new biomaterials, sensors, and other innovative technologies in biology and materials science. We wish our paper could inspire future research in this field.

## Acknowledgments

This work is jointly supported by the National Key R&D Program of China (No.2022ZD0160502), the National Natural Science Foundation of China (No. 61925601, No. 62376276, No. 62276152), and Beijing Nova Program (20230484278).

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

# A  Reconstruction of Full-Atom Geometry

## A.1  Auxilary Loss for Training the AutoEncoder

To better recover the all-atom geometry, we employ an auxilary structural loss similar to the violation loss in Jumper et al. [31] including the supervision on $C_\alpha$ coordinates, bond lengths, and side-chain dihedral angles. First, since the $C_\alpha$ is critical in deciding the global geometry of the peptide, we exert additional loss on its coordinates to distinguish it from other atoms:

$$\mathcal{L}_{CA}(i) = \text{MSE}(\vec{r}_i, \vec{r}_i'), \tag{16}$$

where $\vec{r}_i'$ and $\vec{r}_i$ are the reconstructed and the ground truth coordinates of $C_\alpha$ in node $i$. Next, we implement L1 loss on the bond lengths:

$$\mathcal{L}_{bond}(i) = \sum\nolimits_{b \in \mathcal{B}(i)} |b - b'|/|\mathcal{B}(i)|, \tag{17}$$

where $\mathcal{B}(i)$ includes all chemical bonds in node $i$, and $b'$ denotes the reconstructed bond length. For simplicity, he bonds between residues are included into the bonds of the former residue. Finally, we supervise on the $\chi_1$ to $\chi_4$ side-chain dihedral angles [73]:

$$\mathcal{L}_{angle}(i) = \sum\nolimits_{\chi \in \mathcal{A}(i)} |\chi - \chi'|/|\mathcal{A}(i)|, \tag{18}$$

where $\mathcal{A}(i)$ includes all side-chain dihedral angles in node $i$, $\chi'$ and $\chi$ denotes the reconstructed and the ground truth angles, respectively. The overall auxilary loss for node $i$ is then given by:

$$\mathcal{L}_{aux}(i) = \lambda_{CA}\mathcal{L}_{CA}(i) + \lambda_{bond}\mathcal{L}_{bond}(i) + \lambda_{angle}\mathcal{L}_{angle}(i), \tag{19}$$

where we set $\lambda_{CA} = 1.0, \lambda_{bond} = 1.0, \lambda_{angle} = 0.5$ in our experiments. We find that it is necessary to set $\lambda_{angle}$ with a relatively small value to make the training process stable.

## A.2  Idealization of Local Geometry

Preserving atom instances enhances the modeling of side chain interactions but introduces challenges due to potential twisting in local geometry. While in sequence-structure co-design, samples undergo fast relax by physical force field to ensure a valid local geometry, in binding conformation generation, we use an alignment technique to place the idealized side chains on the generated atom instances. Specifically, the idealized side chain can be represented as at most 4 dihedral angles ($\chi$-angles), treating fragments like phenyl group as rigid bodies. Suppose there are $n_i$ $\chi$-angles and $c_i$ atoms in node $i$, we can define a function to map from the $\chi$-angles with the backbone coordinates $\vec{B}_i \in \mathbb{R}^{4 \times 3}$ to atom instances as $M_i : \mathbb{R}^{n_i} \times \mathbb{R}^{4 \times 3} \to \mathbb{R}^{c_i \times 3}$, which is luckily differentiable [28]. Thus we can optimize $\chi$-angles via gradient descent to minimize the MSE between the coordinates constructed from $\chi$-angles and those generated by the model:

$$\chi_i^* = \arg\min\nolimits_{\chi \in [0, 2\pi]^{n_i}} \|M_i(\chi, \vec{B}_i) - \vec{X}_i\|^2, \tag{20}$$

where $\vec{X}_i \in \mathbb{R}^{c_i \times 3}$ denotes the coordinates of $c_i$ atom instances in node $i$ generated by the model. Now it should be easy to construct an idealized side chain maintaining fidelity to the generated atom instances by $M(\chi_i^*, \vec{B}_i)$. In the experiments of conformation generation on PepBench, such idealization gives slightly better DockQ and $\text{RMSD}_{atom}$ than Rosetta side-chain packing algorithm but with higher efficiency.

# B  Proof of Proposition 3.1

**Proposition 3.1.** *Denote the invariant and equivariant outputs from a scalarization-based E(3)-equivariant GNN as $f(\{\mathbf{h}_i, \vec{\mathbf{x}}_i\})$ and $\vec{f}(\{\mathbf{h}_i, \vec{\mathbf{x}}_i\})$, respectively. With the definition of F in Eq. 7, $\forall g \in E(3)$, we have $f(\{\mathbf{h}_i, F(\vec{\mathbf{x}}_i)\}) = f(\{\mathbf{h}_i, F_g(g \cdot \vec{\mathbf{x}}_i)\})$ and $g \cdot F^{-1}(\vec{f}(\{\mathbf{h}_i, F(\vec{\mathbf{x}}_i)\})) = F_g^{-1}(\vec{f}(\{\mathbf{h}_i, F_g(g \cdot \vec{\mathbf{x}}_i)\}))$, where $F_g$ is derived on the coordinates transformed by g. Namely, the E(3)-equivariance is preserved if we implement the GNN on the standard space and recover the original geometry from the outputs.*

For simplicity, given $F$ derived from a set of coordinates $\{\vec{x}_i\}$ following Eq. 7, we use $F_g$ to indicate the affine transformation derived from $\{g \cdot \vec{x}_i\}$, where $g \in E(3)$. Additionally, we keep the terminology "standard space" for describing the space after the data-specific affine transformation $F$. We begin by proving a key lemma, that is, the E(3)-invariance of distances between two nodes in the standard space converted by $F$:

**Lemma C.1.** *Given two nodes $i$ and $j$ in the geometric graph $\mathcal{G}$, denoting their coordinates as $\vec{x}_i$ and $\vec{x}_j$, their distance in the standard space is E(3)-invariant. Namely, $\forall g \in E(3), \|F(\vec{x}_i) - F(\vec{x}_j)\| = \|F_g(g \cdot \vec{x}_i) - F_g(g \cdot \vec{x}_j)\|$.*

*Proof.* $\forall g \in E(3)$, $g$ can be instantiated as an orthogonal matrix $\boldsymbol{Q} \in O(3)$ (including rotation and reflection), and a translation vector $\vec{t} \in \mathbb{R}^3$. Denoting all coordinates in the geometric graph as $\vec{\boldsymbol{X}} \in \mathbb{R}^{3 \times |\mathcal{G}|}$, and the number of nodes in $\mathcal{G}$ as $n$, we can derive the E(3)-equivariance of the expectation of the coordinates:

$$\mathbb{E}[g \cdot \vec{\boldsymbol{X}}] = \frac{1}{n} \sum_i g \cdot \vec{x}_i = \frac{1}{n} \sum_i (\boldsymbol{Q}\vec{x}_i + \vec{t}) \tag{21}$$

$$= \frac{1}{n} \boldsymbol{Q}(\sum_i \vec{x}_i) + \vec{t} = \boldsymbol{Q}\mathbb{E}[\vec{\boldsymbol{X}}] + \vec{t} \tag{22}$$

$$= g \cdot \mathbb{E}[\vec{\boldsymbol{X}}]. \tag{23}$$

With the E(3)-equivariance of the expectation, it is easy to derive the following equation on the covariance matrix:

$$\mathrm{Cov}(g \cdot \vec{\boldsymbol{X}}, g \cdot \vec{\boldsymbol{X}}) = \frac{1}{n-1}(g \cdot \vec{\boldsymbol{X}} - \mathbb{E}[g \cdot \vec{\boldsymbol{X}}])(g \cdot \vec{\boldsymbol{X}} - \mathbb{E}[g \cdot \vec{\boldsymbol{X}}])^\top \tag{24}$$

$$= \frac{1}{n-1}(g \cdot \vec{\boldsymbol{X}} - g \cdot \mathbb{E}[\vec{\boldsymbol{X}}])(g \cdot \vec{\boldsymbol{X}} - g \cdot \mathbb{E}[\vec{\boldsymbol{X}}])^\top \tag{25}$$

$$= \frac{1}{n-1}(\boldsymbol{Q}(\vec{\boldsymbol{X}} - \mathbb{E}[\vec{\boldsymbol{X}}]))(\boldsymbol{Q}(\vec{\boldsymbol{X}} - \mathbb{E}[\vec{\boldsymbol{X}}]))^\top \tag{26}$$

$$= \frac{1}{n-1}\boldsymbol{Q}(\vec{\boldsymbol{X}} - \mathbb{E}[\vec{\boldsymbol{X}}])(\vec{\boldsymbol{X}} - \mathbb{E}[\vec{\boldsymbol{X}}])^\top \boldsymbol{Q}^\top \tag{27}$$

$$= \boldsymbol{Q}\mathrm{Cov}(\vec{\boldsymbol{X}}, \vec{\boldsymbol{X}})\boldsymbol{Q}^\top. \tag{28}$$

Based on the Cholesky decomposition used in the derivation of $F$, we denote $\mathrm{Cov}(\vec{\boldsymbol{X}}, \vec{\boldsymbol{X}}) = \vec{\boldsymbol{L}}\vec{\boldsymbol{L}}^\top$ and $\mathrm{Cov}(g \cdot \vec{\boldsymbol{X}}, g \cdot \vec{\boldsymbol{X}}) = \vec{\boldsymbol{L}}_g\vec{\boldsymbol{L}}_g^\top$, with which we can immediately derive:

$$\vec{\boldsymbol{L}}_g\vec{\boldsymbol{L}}_g^\top = \boldsymbol{Q}\vec{\boldsymbol{L}}\vec{\boldsymbol{L}}^\top \boldsymbol{Q}^\top. \tag{29}$$

Considering $\boldsymbol{Q}^{-1} = \boldsymbol{Q}^\top$, we further have the following equation:

$$\vec{\boldsymbol{L}}_g^{-\top}\vec{\boldsymbol{L}}_g^{-1} = \boldsymbol{Q}\vec{\boldsymbol{L}}^{-\top}\vec{\boldsymbol{L}}^{-1}\boldsymbol{Q}^\top. \tag{30}$$

Given that $F(\vec{x}) = \vec{\boldsymbol{L}}^{-1}(\vec{x} - \mathbb{E}[\vec{\boldsymbol{X}}])$, we are now ready to prove Lemma C.1 as follows:

$$\|F_g(g \cdot \vec{x}_i) - F_g(g \cdot \vec{x}_j)\| = \|\vec{\boldsymbol{L}}_g^{-1}(g \cdot \vec{x}_i - \mathbb{E}[g \cdot \vec{\boldsymbol{X}}]) - \vec{\boldsymbol{L}}_g^{-1}(g \cdot \vec{x}_j - \mathbb{E}[g \cdot \vec{\boldsymbol{X}}])\| \tag{31}$$

$$= \|\vec{\boldsymbol{L}}_g^{-1}(g \cdot \vec{x}_i - g \cdot \vec{x}_j)\| = \|\vec{\boldsymbol{L}}_g^{-1}\boldsymbol{Q}(\vec{x}_i - \vec{x}_j)\| \tag{32}$$

$$= \sqrt{(\vec{\boldsymbol{L}}_g^{-1}\boldsymbol{Q}(\vec{x}_i - \vec{x}_j))^\top(\vec{\boldsymbol{L}}_g^{-1}\boldsymbol{Q}(\vec{x}_i - \vec{x}_j))} \tag{33}$$

$$= \sqrt{(\vec{x}_i - \vec{x}_j)^\top \boldsymbol{Q}^\top \vec{\boldsymbol{L}}_g^{-\top}\vec{\boldsymbol{L}}_g^{-1}\boldsymbol{Q}(\vec{x}_i - \vec{x}_j)} \tag{34}$$

$$= \sqrt{(\vec{x}_i - \vec{x}_j)^\top \vec{\boldsymbol{L}}^{-\top}\vec{\boldsymbol{L}}^{-1}(\vec{x}_i - \vec{x}_j)} \tag{35}$$

$$= \sqrt{(\vec{\boldsymbol{L}}^{-1}(\vec{x}_i - \vec{x}_j))^\top(\vec{\boldsymbol{L}}^{-1}(\vec{x}_i - \vec{x}_j))} = \|\vec{\boldsymbol{L}}^{-1}(\vec{x}_i - \vec{x}_j)\| \tag{36}$$

$$= \|\vec{\boldsymbol{L}}^{-1}(\vec{x}_i - \mathbb{E}[\vec{\boldsymbol{X}}]) - \vec{\boldsymbol{L}}^{-1}(\vec{x}_j - \mathbb{E}[\vec{\boldsymbol{X}}])\| = \|F(\vec{x}_i) - F(\vec{x}_j)\| \tag{37}$$

$$\square$$

With Lemma C.1, we are able to give the proof of Proposition 3.1 as follows.

*Proof.* A first observation is that to prove Proposition 3.1, we only need to prove the equivariance in the 1-layer case, since the multi-layer case can be decomposed into the 1-layer case by inserting $I = F \circ F^{-1}$ between layers, where $I$ is the identical mapping. Generally, each layer in scalarization-based E(3)-equivariant GNN has the following paradigm:

$$m_{ij} = \phi_m(h_i, h_j, \|\vec{x}_i - \vec{x}_j\|^2, e_{ij}), \tag{38}$$

$$\vec{x}'_i = \vec{x}_i + \sum_{j \in \mathcal{N}(i)} (\vec{x}_i - \vec{x}_j)\phi_x(m_{ij}), \tag{39}$$

$$\vec{h}'_i = \phi_h(h_i, \sum_{j \in \mathcal{N}(i)} m_{ij}), \tag{40}$$

where $\mathcal{N}(i)$ denotes the neighborhood of node $i$, and $\phi_x$ outputs a scalar. Therefore, for the invariant part, we have:

$$f_i(\{h_i, F_g(g \cdot \vec{x}_i)\}) = \phi_h(h_i, \sum_{j \in \mathcal{N}(i)} m_{ij,F_g}) \tag{41}$$

$$= \phi_h(h_i, \sum_{j \in \mathcal{N}(i)} \phi_m(h_i, h_j, \|F_g(g \cdot \vec{x}_i) - F_g(g \cdot \vec{x}_j)\|^2, e_{ij})) \tag{42}$$

$$= \phi_h(h_i, \sum_{j \in \mathcal{N}(i)} \phi_m(h_i, h_j, \|F(\vec{x}_i) - F(\vec{x}_j)\|, e_{ij})) \tag{43}$$

$$= \phi_h(h_i, \sum_{j \in \mathcal{N}(i)} m_{ij,F}) \tag{44}$$

$$= f_i(\{h_i, F(\vec{x}_i)\}) \tag{45}$$

For the equivariant features, recalling $F^{-1}(\vec{x}) = \vec{L}\vec{x} + \mathbb{E}[\vec{X}]$, we have:

$$F_g^{-1}\vec{f}_i(\{h_i, F_g(g \cdot \vec{x}_i)\}) \tag{46}$$

$$= F_g^{-1}(F_g(g \cdot \vec{x}_i) + \sum_{j \in \mathcal{N}(i)} (F_g(g \cdot \vec{x}_i) - F_g(g \cdot \vec{x}_j))\phi_x(m_{ij,F_g})) \tag{47}$$

$$= F_g^{-1}(F_g(g \cdot \vec{x}_i) + \sum_{j \in \mathcal{N}(i)} \vec{L}_g^{-1}Q(\vec{x}_i - \vec{x}_j)\phi_x(m_{ij,F})) \tag{48}$$

$$= F_g^{-1}(\vec{L}_g^{-1}(Q\vec{x}_i + \vec{t} - \mathbb{E}[g \cdot \vec{X}]) + \sum_{j \in \mathcal{N}(i)} \vec{L}_g^{-1}Q(\vec{x}_i - \vec{x}_j)\phi_x(m_{ij,F})) \tag{49}$$

$$= F_g^{-1}(\vec{L}_g^{-1}(Q\vec{x}_i + \vec{t} - g \cdot \mathbb{E}[\vec{X}]) + \sum_{j \in \mathcal{N}(i)} \vec{L}_g^{-1}Q(\vec{x}_i - \vec{x}_j)\phi_x(m_{ij,F})) \tag{50}$$

$$= F_g^{-1}(\vec{L}_g^{-1}(Q\vec{x}_i - Q\mathbb{E}[\vec{X}]) + \sum_{j \in \mathcal{N}(i)} \vec{L}_g^{-1}Q(\vec{x}_i - \vec{x}_j)\phi_x(m_{ij,F})) \tag{51}$$

$$= F_g^{-1}(\vec{L}_g^{-1}Q(\vec{x}_i - \mathbb{E}[\vec{X}]) + \sum_{j \in \mathcal{N}(i)} \vec{L}_g^{-1}Q(\vec{x}_i - \vec{x}_j)\phi_x(m_{ij,F})) \tag{52}$$

$$= F_g^{-1}(\vec{L}_g^{-1}Q(\vec{x}_i - \mathbb{E}[\vec{X}] + \sum_{j \in \mathcal{N}(i)} (\vec{x}_i - \vec{x}_j)\phi_x(m_{ij,F}))) \tag{53}$$

$$= Q(\vec{x}_i - \mathbb{E}[\vec{X}] + \sum_{j \in \mathcal{N}(i)} (\vec{x}_i - \vec{x}_j)\phi_x(m_{ij,F})) + \mathbb{E}[g \cdot \vec{X}] \tag{54}$$

$$= Q(\vec{x}_i + \sum_{j \in \mathcal{N}(i)} (\vec{x}_i - \vec{x}_j)\phi_x(m_{ij,F})) - Q\mathbb{E}[\vec{X}] + g \cdot \mathbb{E}[\vec{X}] \tag{55}$$

$$= Q(\vec{x}_i + \sum_{j \in \mathcal{N}(i)} (\vec{x}_i - \vec{x}_j)\phi_x(m_{ij,F})) + \vec{t} \tag{56}$$

$$= g \cdot (\vec{x}_i + \sum_{j \in \mathcal{N}(i)} (\vec{x}_i - \vec{x}_j)\phi_x(m_{ij,F})) \tag{57}$$

By replacing $g$ with identical element $I$ in $E(3)$, since $F = F_I$, we can immediately derive:

$$F^{-1}\vec{f}_i((\{h_i, F(\vec{x}_i)\}) = (\vec{x}_i + \sum_{j \in \mathcal{N}(i)} (\vec{x}_i - \vec{x}_j)\phi_x(m_{ij,F})), \tag{58}$$

$$g \cdot F^{-1}\vec{f}_i((\{h_i, F(\vec{x}_i)\}) = F_g^{-1}\vec{f}_i(\{h_i, F_g(g \cdot \vec{x}_i)\}), \tag{59}$$

which concludes Proposition 3.1. □

## D  Algorithm for Training and Sampling

We present the pseudo codes for training in Algorithm 1 amd sampling in Algorithm 2.

---

**Algorithm 1** Training Algorithm of PepGLAD

---

**input**  geometric data of protein-peptide complexes $\mathcal{S}$
**output**  encoder $\mathcal{E}_\phi$, decoder $\mathcal{D}_\xi$, denoising network $\epsilon_\theta$

 1: **function** TrainAutoEncoder($\mathcal{S}$)
 2:     Initialize $\mathcal{E}_\phi, \mathcal{D}_\xi$
 3:     **while** $\phi, \xi$ have not converged **do**
 4:         Sample $(\mathcal{G}_p, \mathcal{G}_b) \sim \mathcal{S}$
 5:         $\tilde{\mathcal{G}}_p \leftarrow \text{mask}(\mathcal{G}_p)$                                                    {Mask 25% Residues}
 6:         $\{(\boldsymbol{\mu}_i, \boldsymbol{\sigma}_i, \vec{\boldsymbol{\mu}}_i, \vec{\boldsymbol{\sigma}}_i)\} \leftarrow \mathcal{E}_\phi(\tilde{\mathcal{G}}_p, \mathcal{G}_b)$            {Encoding}
 7:         $\{(\boldsymbol{\epsilon}_i, \vec{\boldsymbol{\epsilon}}_i)\} \sim \mathcal{N}(\mathbf{0}, \boldsymbol{I})$                        {Reparameterization}
 8:         $\mathcal{G}_z \leftarrow \{(\boldsymbol{\mu}_i + \boldsymbol{\epsilon}_i \odot \boldsymbol{\sigma}_i, \vec{\boldsymbol{\mu}}_i + \vec{\boldsymbol{\epsilon}}_i \odot \vec{\boldsymbol{\sigma}}_i)\}$
 9:         $\mathcal{G}'_p \leftarrow \mathcal{D}_\xi(\mathcal{G}_z, \mathcal{G}_b)$                                        {Decoding}
10:         $\mathcal{L}_{AE} = \sum_{i \in \mathcal{G}_p}(\mathcal{L}_{recon}(i) + \mathcal{L}_{KL}(i))/|\mathcal{G}_p|$
11:         $\phi, \xi \leftarrow \text{optimizer}(\mathcal{L}_{AE}; \phi, \xi)$
12:     **end while**
13:     **return** $\mathcal{E}_\phi, \mathcal{D}_\xi$
14: **end function**
15:
16: **function** TrainLatentDiffusion($\mathcal{E}_\phi, \mathcal{D}_\xi, \mathcal{S}$)
17:     Initialize $\epsilon_\theta$
18:     **while** $\theta$ have not converged **do**
19:         Sample $(\mathcal{G}_p, \mathcal{G}_b) \sim \mathcal{S}$
20:         $\{(\boldsymbol{\mu}_i, \boldsymbol{\sigma}_i, \vec{\boldsymbol{\mu}}_i, \vec{\boldsymbol{\sigma}}_i)\} \leftarrow \mathcal{E}_\phi(\mathcal{G}_p, \mathcal{G}_b)$                    {Encoding}
21:         $\mathcal{G}_z^0 \leftarrow \{(\boldsymbol{\mu}_i, \vec{\boldsymbol{\mu}}_i)\}$
22:         $F \leftarrow \text{affine}(\mathcal{G}_b)$                                            {Affine Transformation}
23:         $\mathcal{G}_z^0, \mathcal{G}_b \leftarrow F(\mathcal{G}_z^0), F(\mathcal{G}_b)$                                {Standard Geometry}
24:         $\{(\boldsymbol{z}_i^0, \vec{\boldsymbol{z}}_i^0)\} \leftarrow \mathcal{G}_z^0$
25:         $t \sim \mathbf{U}(1, T), \{(\boldsymbol{\epsilon}_i, \vec{\boldsymbol{\epsilon}}_i)\} \sim \mathcal{N}(\mathbf{0}, \boldsymbol{I})$
26:         $\mathcal{G}_z^t \leftarrow \{\sqrt{\bar{\alpha}^t}[\boldsymbol{z}_i^0, \vec{\boldsymbol{z}}_i^0] + (1 - \bar{\alpha}^t)[\boldsymbol{\epsilon}_i, \vec{\boldsymbol{\epsilon}}_i]\}$
27:         $\mathcal{L}_{LDM}^t = \sum_i \|[\boldsymbol{\epsilon}_i, \vec{\boldsymbol{\epsilon}}_i] - \epsilon_\theta(\mathcal{G}_z^t, \mathcal{G}_1, t)[i]\|^2/|\mathcal{G}_z^t|$
28:         $\theta \leftarrow \text{optimizer}(\mathcal{L}_{LDM}^t; \theta)$
29:     **end while**
30:     **return** $\epsilon_\theta$
31: **end function**
32:
33: $\mathcal{E}_\phi, \mathcal{D}_\xi \leftarrow \text{TrainAutoEncoder}(\mathcal{S})$
34: Fix parameters $\phi$ and $\xi$
35: $\epsilon_\theta \leftarrow \text{TrainLatentDiffusion}(\mathcal{E}_\phi, \mathcal{D}_\xi, \mathcal{S})$
36: **return** $\mathcal{E}_\phi, \mathcal{D}_\xi, \epsilon_\theta$

---

## E  Data Preparation

We show details for constructing the datasets used in our paper here. Further statistics are presented in Table 5 and distribution of peptide lengths in Figure 5.

### E.1  Unsupervised Data from Protein Fragments (ProtFrag)

We exploit unsupervised data from monomer proteins to enrich the training of the autoencoder. Specifically, we first extract all single chains from the Protein Data Bank (PDB) before December 8th, 2023, and remove the duplicated chains on a sequence identity threshold of over 90%. Then, for each chain, we extract fragments satisfying the following criteria:

---

**Algorithm 2** Sampling Algorithm of PepGLAD

---

**input** decoder $\mathcal{D}_\xi$, denoising network $\epsilon_\theta$, binding site $\mathcal{G}_b$
**output** peptide $\mathcal{G}_p$

1: $F \leftarrow \text{affine}(\mathcal{G}_b)$         {Affine Transformation}
2: $\mathcal{G}_b \leftarrow F(\mathcal{G}_b)$         {Standard Geometry}
3: $\{(\boldsymbol{z}_i^T, \vec{\boldsymbol{z}}_i^T)\} \sim \mathcal{N}(\boldsymbol{0}, \boldsymbol{I})$
4: **for** $t$ in $T, T-1, \cdots, 1$ **do**
5:      $\mathcal{G}_z^t \leftarrow \{(\boldsymbol{z}_i^t, \vec{\boldsymbol{z}}_i^t)\}$         {Latent Denoising Loop}
6:      $\vec{\boldsymbol{u}}_i^t \leftarrow \boldsymbol{z}_i^t, \vec{\boldsymbol{z}}_i^t$
7:      $\boldsymbol{\varepsilon}_i = [\boldsymbol{\epsilon}_i, \vec{\boldsymbol{\epsilon}}_i] \sim \mathcal{N}(\boldsymbol{0}, \boldsymbol{I})$
8:      $\vec{\boldsymbol{u}}_i^{t-1} \leftarrow \frac{1}{\sqrt{\alpha^t}}(\vec{\boldsymbol{u}}_i^t - \frac{\beta^t}{\sqrt{1-\bar{\alpha}^t}} \epsilon_\theta(\mathcal{G}_z^t, \mathcal{G}_b, t)[i]) + \beta_t \boldsymbol{\varepsilon}_i$
9:      $[\boldsymbol{z}_i^{t-1}, \vec{\boldsymbol{z}}_i^{t-1}] \leftarrow \vec{\boldsymbol{u}}_i^{t-1}$
10: **end for**
11: $\mathcal{G}_z \leftarrow \{(\boldsymbol{z}_i^0, \vec{\boldsymbol{z}}_i^0)\}$
12: $\mathcal{G}_z, \mathcal{G}_b \leftarrow F^{-1}(\mathcal{G}_z), F^{-1}(\mathcal{G}_b)$         {Data Geometry}
13: $\mathcal{G}_p \leftarrow \mathcal{D}_\xi(\mathcal{G}_z, \mathcal{G}_b)$         {Decoding}
14: **return** $\mathcal{G}_p$

---

1. **Length**: The fragment should consist of 4 to 25 residues.

2. **Balanced constitution**: No single amino acid should constitute more than 25% of the fragment; Hydrophobic amino acids should comprise less than 45% of the fragment, with charged amino acids accounting for 25% to 45%.

3. **Isolated Stability**: Instability [22] should be below 40; Considering the surrounding amino acids as interaction partners, the fragment should have a buried surface area (BSA) above $400\text{Å}^2$ [10], with a relative BSA above 20%.

We use FreeSASA [47] to calculate the surface area of fragments. Let $\text{SA}_{bound}$ represent the surface area of the isolated fragment when considering surrounding amino acids, and $\text{SA}_{unbound}$ represent the surface area when not considering them. The buried surface area is then calculated as $\text{BSA} = \text{SA}_{unbound} - \text{SA}_{bound}$, and the relative BSA is calculated as $\text{BSA}_{rel} = \text{BSA}/\text{SA}_{unbound}$. In total, we obtain 70,645 fragments meeting these criteria.

### E.2 Construction of Our PepBench

Here we illustrate the details for constructing the supervised dataset (PepBench) used in our paper. Similar to literature [65], we also exploits available data from the Protein Data Bank (PDB) [5]. We first extract all dimers in PDB deposited before December 8th, 2023, and filter out the complexes with a receptor longer than 30 residues and a ligand between 4 to 25 residues, which aligns with Tsaban et al. [59]. Peptides with lengths in this range are more relevant to practical applications such as drug discovery, as they exhibit favorable biochemical properties [49]. Then we remove the duplicated complexes with the criterion that both the receptor and the peptide has a sequence identity over 90% [56], after which 6105 non-redundant complexes are obtained. We use MMseqs2 for clustering based on sequence identity:

```
# create database from the sequences
mmseqs create seqs.fasta database
# clustering with sequence identity above 90%
mmseqs cluster database database_clusters results --min-seq-id 0.9 -c 0.95 --cov-mode 1
```

To achieve the cross-target generalization test, we utilize the large non-redundant dataset (LNR) introduced by Tsaban et al. [59] as the test set, which is curated by domain experts. LNR originally includes 96 protein-peptide complexes. We obtain 93 complexes after excluding the ones with non-canonical amino acids. We then cluster the LNR along with the PDB data by receptor with a sequence identity threshold of over 40%. Subsequently, we remove the complexes sharing the same clusters with those from the test set and those including non-canonical amino acids in the peptides. Finally, the remaining data are randomly split based on clustering results into training and validation sets. The characteristics of different splits are presented in Table 5. In addition, the binding site

contains residues on the receptor within 10Å distances to the peptide, where the distance between two residues is measured by the distance between their $C_\beta$ coordinates.

### E.3 Split of PepBDB

Similar to the split of PepBench, we use MMseqs2 for clustering and randomly split the data into training, validation, and test sets based on the clustering results. For the test set, we randomly select one protein-peptide complex in each cluster to avoid redundancy.

Table 5: Statistics of the constructed datasets.

| Split | #entry | #cluster | source |
|---|---|---|---|
| PepBench (Training) | 4,157 | 952 | PDB [5] |
| PepBench (Validation) | 114 | 50 | PDB [5] |
| PepBench (Test) | 93 | 93 | LNR [59] |
| PepBDB (Training) | 8,434 | 1,617 | PepBDB [65] |
| PepBDB (Validation) | 370 | 95 | PepBDB [65] |
| PepBDB (Test) | 190 | 190 | PepBDB [65] |
| ProtFrag | 70,645 | - | monomers in PDB |

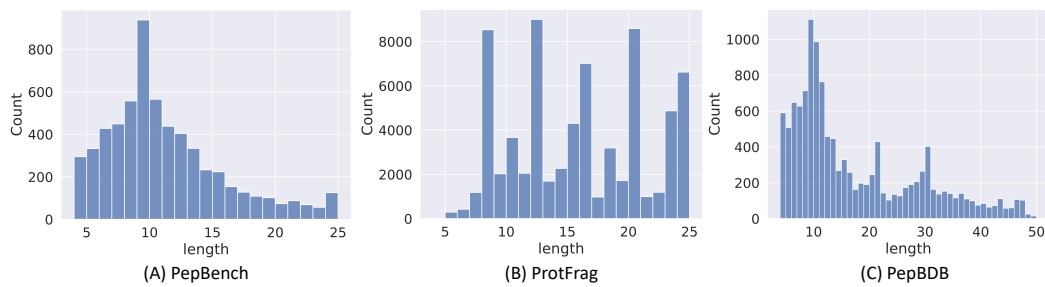

Figure 5: Peptide (or protein fragment) length distribution of three datasets.

## F E(3)-Invariant Latent Space

We found that the E(3)-equivariant latent vectors were significant, which can convey sufficient geometric information. If we use E(3)-invariant latent space without these geometric latent vectors, the reconstruction ability (RMSD, DockQ) of the VAE deteriorates significantly (Table 6). It is reasonable since E(3)-invariant latent space lacks geometric interactions with the pocket atoms, leading to difficulties in reconstructing the full-atom structures on the binding site.

Table 6: Comparison of reconstruction ability of variational autoencoders with E(3)-equivariant latent space and E(3)-invariant latent space.

| Latent Space | AAR↑ | RMSD↓ | DockQ↑ |
|---|---|---|---|
| E(3)-equivariant | 95.1% | 0.79 | 0.898 |
| E(3)-invariant | 93.4% | 1.75 | 0.823 |

## G Guidance on Sequence Orders for Sampling

Peptides consist of linearly connected amino acids, which exert constraints on the 3D geometry. Specifically, residues adjacent in the sequence should also be close in the structure, since they are connected by a peptide bond. However, 3D graphs are unordered and do not incorporate such induct bias, which means the generated nodes might have arbitrary permutation on sequence orders. To tackle this problem, we take inspiration from classifier-guided diffusion [15], which adds the gradient

of a classifier to the denoising outputs to guide the generative diffusion process towards the subspace where the classifier gives high confidence. We utilize an empirical classifier $p(1|\{\vec{z}_i^t\})$ defined as follows:

$$p(1|\{\vec{z}_i^t\}) = \exp(-\sum_{\mathcal{P}(i)-\mathcal{P}(j)=1} E(\|\vec{z}_i^t - \vec{z}_j^t\|)), \tag{60}$$

$$E(d) = \begin{cases} d - (\mu_d + 3\sigma_d), & d > \mu_d + 3\sigma_d, \\ (\mu_d - 3\sigma_d) - d, & d < \mu_d - 3\sigma_d, \\ 0, & \text{otherwise}, \end{cases} \tag{61}$$

where $\mu_d$ and $\sigma_d$ are the mean and variance of the distances of adjacent residues in the latent space measured from the training set. With $p(1|\{\vec{z}_i^t\})$, we are able to assign an arbitrary permutation on sequence orders to the nodes, and steer the sampling procedure towards the desired subspace conforming to $\mathcal{P}$, since this classifier gives higher confidence if the adjacent (defined by $\mathcal{P}$) residues are within reasonable distances aligning with the statistics from the training set. In particular, the coordinate denoising outputs are refined as follows:

$$\vec{\epsilon}_i^t = \vec{\epsilon}_\theta(\mathcal{G}_z^t, \mathcal{G}_b, t)[i] - \lambda\sqrt{1 - \bar{\alpha}^t}\nabla_{\vec{z}_i^t}\log p(1|\{\vec{z}_i^t\}), \tag{62}$$

where $\lambda$ adjusts the weight of the guidance. Besides the constraints on the distance between adjacent residues, we can also include guidance on avoiding clashes between non-adjacent residues by defining the following energy term:

$$C(d) = \begin{cases} \mu_d - d, & d < \mu_d, \\ 0, & \text{otherwise}, \end{cases} \tag{63}$$

Subsequently, we just need to revise the empirical classifier as:

$$p(1|\{\vec{z}_i^t\}) = \exp(-\sum_{\mathcal{P}(i)-\mathcal{P}(j)=1} E(\|\vec{z}_i^t - \vec{z}_j^t\|) - \sum_{\mathcal{P}(i)-\mathcal{P}(j)\neq 1} C(\|\vec{z}_i^t - \vec{z}_j^t\|) - \sum_{i\in\mathcal{G}_z, j\in\mathcal{G}_b} C(\|\vec{z}_i^t - \vec{r}_j\|), \tag{64}$$

where $\vec{r}_j$ is the $\texttt{C}_\alpha$ coordinate of node $j$ in the binding site. We observe a slight improvement upon including the clash energy term. Nevertheless, the guidance is only a small technical trick with minor enhancement on the performance as shown in Table 7.

Table 7: Results on binding conformation generation with and without guidance on sequence orders.

| Guidance | RMSD$_{C_\alpha}\downarrow$ | RMSD$_{\text{atom}}\downarrow$ | DockQ$\uparrow$ |
|---|---|---|---|
| w/ | 4.09 | 5.30 | 0.592 |
| w/o | 4.10 | 5.34 | 0.582 |

# H  Metrics for Evaluating Sequence-Structure Co-Design

## H.1  Why not Amino Acid Recovery (AAR)

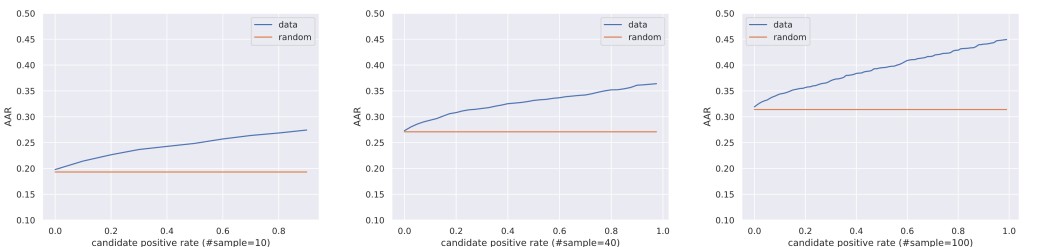

Figure 6: Best amino acid recovery (AAR) on samples constructed according to specified positive ratios. Each point contains the averaged result across 102 receptors.

While the amino acid recovery (AAR) is widely used in antibody design [33, 34, 44], we find it not informative enough for evaluating generative models on learning the distribution of binding peptides, due to the vast and highly diverse solution space. To elucidate this, we conducted an analysis of AAR using a sequence dataset from Xia et al. [67] including 328 receptors and 600k peptides with binary binding labels, from which we filter out 102 receptors with well-explored solution space (*i.e.* with at least 500 binders). For each receptor, we randomly select one binder as the reference and sample $N$ candidates according to a specified positive ratio $r$, resembling the scenario of evaluating generative models for peptide design. For example, setting $r = 0.1$ and $N = 100$ involves sampling 10 binders and 90 non-binders to construct the candidates. Then we compute the best AAR of these candidates with respect to the reference as the result. This process is repeated for 10 times per receptor, and the results are averaged across different receptors, which can be interpreted as the evaluation score of AAR on a generative model that can generates candidates with a positive ratio of $r$. We select $N = 10, 40, 100$ and enumerate the choices of $r$ to derive the relation plot in Figure 6, where the results of a random sequence generator is also included for comparison. It can be derived that the gap of AAR between the worst model ($r = 0.0$) and the best model ($r = 1.0$) is insignificant. Furthermore, all models, including the best model ($r = 1.0$) which always produces positive samples, exhibit performance akin to the random sequence generator, with consistent trends regardless of the choice of $N$. We attribute this to the vast solution space of peptide design, where evaluating with dozens of candidates relative to a single reference is unreliable. In other words, **achieving a high AAR is improbable since the model would need to fortuitously explore the subspace around the reference, which is arbitrary, on every test receptor; Conversely, a low AAR does not necessarily denote a poor generative model, as the model may be exploring distinct solutions from the single reference.** Moreover, we calculate the Spearman correlation between the receptor-level best AAR and the positive ratio $r$ of the candidates, yielding 0.23, 0.22 and 0.24 for $N = 10, 40,$ and 100, respectively, indicating very weak correlation. Based on this analysis, we assert that AAR is unsuitable for evaluating target-specific peptide design. The comparison of AAR and success rates on PepBench (Table 8) also indicates that models with similar AAR can have distinct success rates.

Table 8: Amino acid recovery and success rates of different models on PepBench.

| Model | AAR | Success ($\Delta G < 0$) |
|---------|-------|--------------------------|
| HSRN | 35.8% | 10.46% |
| DiffAb | 37.1% | 49.87% |
| PepGLAD | 36.7% | 55.97% |

## H.2 "Unsupervised" Consistency

While it is common to evaluate consistency by comparing generated structures with AF2-predicted structures [7], such a "supervision-based" method suffers from severe limitations in the situation where AF2 [31] fails to achieve an acceptable performance. As highlighted not only in Tsaban et al. [59] but also demonstrated by our experiments, even state-of-the-art models like AF2 struggles to consistently produce high-quality structures of protein-peptide complexes. Our findings reveal that only 36% of test samples could be accurately predicted within a 5Å RMSD (considered as near-native conformation) by AF2, indicating even the ground truth can only achieve 36% success rates if we use such supervison-based consistency for evaluation, making such evaluation not reliable in the domain of peptide design.

In light of this limitation, we propose an "unspervised" evaluation framework to assess consistency, that is, the statistical association between the clustering results of sequences and structures. Theoretically, this serves the necessary condition for true consistency. Namely, if a model truly captures the consistency between sequence and structure, it will necessarily achieve a high score on the proposed metric. Conversely, if a model fails to attain a high score on the proposed metric, it is not possible to capture true consistency.

In our experiments, we found that our proposed consistency metric effectively distinguishes non-consistent modeling methods, such as HSRN, which tend to produce disparate sequences while sharing identical structures.

### H.3 Implememtation and Elaboration on Metrics

Indeed, evaluating generative models comprehensively is crucial, requiring assessment from multiple perspectives. Generally, these evaluations can be categorized into two main aspects: diversity and fidelity to the desired distribution. Regarding diversity, we take inspiration from Yim et al. [72] and quantify it with the number of unique clusters relative to the number of candidates. This metric provides insight into the variety and richness of the generated samples. For fidelity, the primary focus should be the binding affinity, for which we adopt the physical energy from Rosetta [2] since it is widely used in various domains and exhibit robust generalization capability [1, 37, 64]. Further, considering the dependency between sequence and structure, we propose the consistency metric, which is critical for distinguishing whether the generative model is capturing the 1D&3D joint distribution and thereby truly facilitating "co-design". We have also discussed the reason for implementing the consistency metric with the "unsupervised" fashion in the above section. Below, we outline the implementation of these metrics.

**Diversity**    We hierarchically cluster the sequences and the structures with aligning score [36] and RMSD of $C_\alpha$, respectively. In particular, we implement the aligning score using Biopython [11] with BLOSUM62 matrix [24] and Needleman-Wunsch algorithm [50]. The thresholds for clustering is similarity above 0.6 and RMSD below 4.0 for sequence and structure, respectively. We provide the python codes below for clearer presentation:

```python
from math import sqrt
from typing import List
from Bio.Align import substitution_matrices, PairwiseAligner

import numpy as np

def align_score(sequence_A, sequence_B):
    # load matrix
    sub_matrice = substitution_matrices.load('BLOSUM62')
    aligner = PairwiseAligner()
    aligner.substitution_matrix = sub_matrice
    # align
    alns = aligner.align(sequence_A, sequence_B)
    best_aln = alns[0]
    aligned_A, aligned_B = best_aln
    # normalize
    base_A = aligner.score(sequence_A, sequence_A)
    base_B = aligner.score(sequence_B, sequence_B)
    base = sqrt(base_A * base_B)
    similarity = aligner.score(sequence_A, sequence_B) / base
    return similarity

def seq_diversity(seqs: List[str], th: float=0.4) -> float:
    '''
        th: sequence distance (1 - similarity)
    '''
    dists = []
    for i, sequence_A in enumerate(seqs):
        dists.append([])
        for j, sequence_B in enumerate(seqs):
            dists[i].append(1 - align_score(sequence_A, sequence_B))
    dists = np.array(dists)
    Z = linkage(squareform(dists), 'single')
    cluster = fcluster(Z, t=th, criterion='distance')
    return len(np.unique(cluster)) / len(seqs)

def struct_diversity(structs: np.ndarray, th: float=4.0) -> float:
    '''
        structs: N*L*3, alpha carbon coordinates
        th: threshold for clustering (distance < th)
    '''
    ca_dists = np.sum((structs[:, None] - structs[None, :]) ** 2, axis=-1)
    rmsd = np.sqrt(np.mean(ca_dists, axis=-1))
    Z = linkage(squareform(rmsd), 'single')
    cluster = fcluster(Z, t=th, criterion='distance')
    return len(np.unique(cluster)) / structs.shape[0]
```

Denoting the diversity of the sequences and the structures as $\text{Div}_{seq}$ and $\text{Div}_{struct}$, respectively, we calculate the co-design diversity as $\sqrt{\text{Div}_{seq}\text{Div}_{struct}}$.

**Consistency** The clustering process in the calculation of diversity will assign each sequence and each structure with a clustering label. The labels on the sequences and those on the structures can be regarded as two nominal variables. Since similar sequences should produce similar structures, these two variables should be highly correlated if the model really learns the joint distribution. Naturally, we quantify the consistency via Cramér's V [12] association between these two variables, with 1.0 indicating perfect association, and 0.0 means no association.

**ΔG** We use Rosetta [2] to calculate the binding energy with the "ref2015" score function. Both the candidates and the references first endure the fast relax protocol in Rosetta to correct atomic clashes at the interface before the computation of binding energy. We use the implementation in pyRosetta (*i.e.* the python version of Rosetta), which is borrowed from Luo et al. [44]:

```python
import pyrosetta
from pyrosetta.rosetta import protocols
from pyrosetta.rosetta.protocols.relax import FastRelax
from pyrosetta.rosetta.core.pack.task import TaskFactory
from pyrosetta.rosetta.core.pack.task import operation
from pyrosetta.rosetta.core.select import residue_selector as selections
from pyrosetta.rosetta.core.select.movemap import MoveMapFactory, move_map_action
from pyrosetta.rosetta.core.scoring import ScoreType
from pyrosetta.rosetta.protocols.analysis import InterfaceAnalyzerMover

pyrosetta.init(' '.join([
    '-mute', 'all',
    '-use_input_sc',
    '-ignore_unrecognized_res',
    '-ignore_zero_occupancy', 'false',
    '-load_PDB_components', 'false',
    '-relax:default_repeats', '2',
    '-no_fconfig',
    '-use_terminal_residues', 'true',
    '-in:file:silent_struct_type', 'binary'
]))

class RelaxRegion(object):  # Fast Relax

    def __init__(self, max_iter=1000):
        super().__init__()
        self.scorefxn = pyrosetta.create_score_function('ref2015')
        self.fast_relax = FastRelax()
        self.fast_relax.set_scorefxn(self.scorefxn)
        self.fast_relax.max_iter(max_iter)

    def __call__(self, pdb_path, peptide_chain):
        pose = pyrosetta.pose_from_pdb(pdb_path)

        tf = TaskFactory()
        tf.push_back(operation.InitializeFromCommandline())
        tf.push_back(operation.RestrictToRepacking())   # Only allow residues to repack.
  No design at any position.

        gen_selector = selections.ChainSelector(chain)
        nbr_selector = selections.NeighborhoodResidueSelector()
        nbr_selector.set_focus_selector(gen_selector)
        nbr_selector.set_include_focus_in_subset(True)
        subset_selector = nbr_selector

        prevent_repacking_rlt = operation.PreventRepackingRLT()
        prevent_subset_repacking = operation.OperateOnResidueSubset(
            prevent_repacking_rlt,
            subset_selector,
            flip_subset=True,
        )
        tf.push_back(prevent_subset_repacking)

        fr = self.fast_relax
        pose = original_pose.clone()

        mmf = MoveMapFactory()
        mmf.add_bb_action(move_map_action.mm_enable, gen_selector)
        mmf.add_chi_action(move_map_action.mm_enable, subset_selector)
        mm  = mmf.create_movemap_from_pose(pose)

        fr.set_movemap(mm)
        fr.set_task_factory(tf)
        fr.apply(pose)
```

```
        return pose

def pyrosetta_interface_energy(pdb_path, receptor_chains, ligand_chains): # binding
    energy
    pose = pyrosetta.pose_from_pdb(pdb_path)
    interface = ''.join(ligand_chains) + '_' + ''.join(receptor_chains)
    mover = InterfaceAnalyzerMover(interface)
    mover.set_pack_separated(True)
    mover.apply(pose)
    return pose.scores['dG_separated']
```

**Success**  Since a negative $\Delta G$ value typically indicates a potential for binding, we report the ratio of all generated candidates that satisfy this threshold. Moreover, candidates with $\Delta G < 0$ usually are at least physically valid (*i.e.* without obvious atomic clash), thus it is meaningful to use this success rate to evaluation the generative ability of the models.

# I   Experiment Details

## I.1   Implementation of PepGLAD

We train PepGLAD on a GPU with 24G memory with AdamW optimizer. For the autoencoder, we train for 60 epochs with dynamic batches, ensuring that the total number of edges (proportional to the square of the number of nodes) remains below 60,000. The initial learning rate is $10^{-4}$ and decays by 0.8 if the loss on the validation set has not decreased for 3 consecutive epochs. Regarding the diffusion model, we train for 500 epochs with the same batching strategy as for the autoencoder. The learning rate is $10^{-4}$ and decay by 0.6 if the loss has not decreased for 3 consecutive validations, where the validation is conducted every 10 epochs. In the experiment of binding conformation generation, we only use the supervised dataset, thus we extend the training epochs for the autoencoder and the diffusion model to 500 and 1000, respectively. Consequently, the patience of learning rate decay is extended to 15 epochs for training the autoencoder. We keep other settings unchanged. The hyperparameters of PepGLAD used in our experiments are provided in Table 9.

Table 9: Hyperparameters of PepGLAD in sequence-structure codesign and binding conformation generation.

| Name | Value | | Description |
| --- | --- | --- | --- |
| | codesign | conformation | |
| Variational AutoEncoder | | | |
| embed size | 128 | 128 | Size of embeddings for residue types. |
| hidden size | 128 | 128 | Size of hidden layers. |
| $h$ | 8 | - | Size of the latent variable for residue types in the sequences. |
| layers | 3 | 3 | Number of layers. |
| $\lambda_1$ | 0.1 | 0.0 | The weight of KL divergence on the sequence. |
| $\lambda_2$ | 0.5 | 0.5 | The weight of KL divergence on the structure. |
| $\lambda_{CA}$ | 1.0 | 1.0 | The weight of $C_\alpha$ loss in $\mathcal{L}_{aux}$. |
| $\lambda_{bond}$ | 1.0 | 1.0 | The weight of bond loss in $\mathcal{L}_{aux}$. |
| $\lambda_{angle}$ | 0.5 | 0.5 | The weight of side-chain dihedral angle loss in $\mathcal{L}_{aux}$. |
| Latent Diffusion Model | | | |
| hidden size | 128 | 128 | Size of hidden layers in the denoising network. |
| layers | 3 | 3 | Number of layers. |
| steps | 100 | 100 | Number of diffusion steps. |

## I.2   Implementation of the Baselines

For **HSRN** [30], **dyMEAN** [34], and **DiffAb** [44], we directly integrate their official implementation into the same training framework as our PepGLAD, and adjust the batch size, learning rate, and training epochs to obtain the optimal performance, which we present in Table 10.

We outline the implementation of other baselines below:

Table 10: Hyperparamenters for training the baselines.

| Model | Batch Size | Learning Rate | Epoch |
|--------|------------|---------------|-------|
| HSRN | 8 | 1.0e-4 | 50 |
| dyMEAN | 32 | 1.0e-4 | 100 |
| DiffAb | 32 | 1.0e-4 | 50 |

**RFDiffusion** [64] We follow the official instruction to randomly select 20% of residues on the binding site as "hotspots", and generate the backbone via diffusion followed by cycles of inverse folding with ProteinMPNN [14] and full-atom structural refining with the officially provided Rosetta protocol.

**AnchorExtension** [26] It is implemented with the Rosetta suite, thus we resort to the official release of the pipeline protocols for docking and optimizing. We use the default parameters and generate 10 candidates for each receptor due to its limitation of efficiency. We randomly pick one peptide in the training set with the same length as the reference peptide as the initial motif for docking.

**FlexPepDock** [42] We follow its official tutorial to implement this baseline in the C++ version of Rosetta.

**AlphaFold2** [31] We borrow the results from [59], which explores two strategy to use AlphaFold2 on peptide conformation generation, including modeling the receptor and the peptide as separate chains or link them together with long loops. The results contain a total of 10 candidates for each receptor, with 5 from the separate strategy and 5 from the linked strategy.

