# OpenReview forum: "Full-Atom Peptide Design with Geometric Latent Diffusion"
_NeurIPS.cc/2024/Conference — NeurIPS 2024 poster_

### Official Review · Reviewer_XM8c · 2024-07-10

**Soundness:** 2
**Presentation:** 3
**Contribution:** 3
**Rating:** 6
**Confidence:** 3

**Summary:**

The paper presents a new diffusion model for generating peptide binders given protein pockets. It also presents a new benchmark dataset, created by selecting examples from PDB and ensuring sequence dissimilarity between training and test data.

**Strengths:**

Thank you for constructing a benchmark dataset for this task with a proper train/test split. This is overlooked in some related work. Is the dataset available to download?

Notation is clearly set out and there is a good survey of related work.

There are sensible ablations and comparisons to baselines, and the presented model performs well relative to these.

**Weaknesses:**

In the introduction and abstract, please state clearly that PepGLAD needs to be given the binding site (not the whole protein structure).

Line 24 is hard to understand: ‘The key of peptide design …’.  It seems to say that strong binding requires the peptide to adopt a compact, inflexible shape, but I do not understand why.

The argument for the affine transformation is unclear.  I could not decipher line 48 ‘These variances define divergent target distributions of Gaussian…’.  Plenty of methods (AlphaFold, RFDiffusion, GeoDiff, …) are able to predict or design elongated protein and molecule shapes without any such transform, and here it should be easy because the binding site shape is given as a conditioning input. Surely a model can learn that the binder it generates should approximately match the shape of the binding site, even if the binding site is far from spherical?

I think that proposition 3.1 could be stronger. Please check, but I think that if $x$ is rotated, then $L$ undergoes the same rotation, and hence $F_g(g.x)=F(x)$. Thus, $f(h, F(x))$ is invariant and $F^{-1}\overrightarrow{f}(h, F(x))$ is equivariant even if $f$ and  $\overrightarrow{f}$ are not.

I am surprised that this transformation improves performance.  The score model in the transformed space now has to learn that bonds in different orientations should have different lengths, inferring from the inter-residue distances of the binding site how much each dimension is compressed or stretched.

**Questions:**

Why do you need to generate all-atom designs? Do you think it could work just as well to generate C-alpha coordinates and discrete residue types? Mixed discrete/continuous diffusion models have shown some success for small-molecule drug design (DiffSBDD) and crystal design (MatterGen).

Line 223: does ‘filter out’ mean ‘keep’ or ‘discard’?

The dataset seems small for deep learning and for the complexity of the task.   How could it be augmented?

Table 3: what counts as a good RMSD, for practical purposes? Are all the methods given the same inputs, or are some doing blind docking?

Equation (3): how is the MSE on full-atom structures defined when the residue type is wrong?

**Limitations:**

Like all peptide design in silico, this is hamstrung by lack of good in silico evaluation: dG values from Rosetta are not reliable estimates of real binding affinity. Ideally there would be some wet-lab evaluation of the generated designs.

The method requires binding site to be identified before peptide binders are designed.

---

> ### Author Rebuttal · Authors · 2024-08-06
>
> Thanks for your thoughtful reviews!
>
> > W1: In the intro/abs, please state clearly that PepGLAD needs to be given the binding site.
>
> Thanks for the suggestion! We will state it clearly in the revision.
> > W2: Line 24 is hard to understand: ‘The key of peptide design …’.
>
> Sorry for the confusion. First, it means that to achieve strong binding, the peptide should form compact interactions with the pocket, since dense secondary bonding (e.g. hydrogen bond) contributes to the overal binding affinity. Second, the binding conformation of the peptide is largely affected by the pocket. Free peptides are usually flexible [a], whereas it may adopt certain modes upon binding to form denser interactions.
>
> [a] "The X‐Pro peptide bond as an NMR probe for conformational studies of flexible linear peptides." Biopolymers: Original Research on Biomolecules 15.10 (1976).
> > W3: The argument for the affine transformation is unclear. I could not decipher line 48 ‘These variances define divergent target distributions of Gaussian…’.
>
> Thanks for the comment!
>
> The need for affine transformation arises from the unique challenges associated with peptides compared to general proteins and small molecules:
>
> 1. **Irregular Conformation Scales**: Small molecules have minor scale variances, while protein spread is largely determined by residue number, regularized by the radius of gyration [b]. Conformations for peptides with the same length can vary widely, from linearly extended to compact ball-like structures, making normalization with a shape prior particularly crucial.
>
> 2. **Dataset Size and Generalization**:  The smaller peptide dataset necessitates a well-defined geometric diffusion space for better generalization. Our affine-based normalization in PepGLAD achieves consistent and accurate RMSDs, as shown in Figure 4, compared to DiffAb's Gaussian normalization, which often results in outliers with high RMSDs.
>
> [b] "Radius of gyration as an indicator of protein structure compactness." Molecular Biology 42 (2008).
> > W4: I think that proposition 3.1 could be stronger. Please check, I think that if $x$ is rotated, then $L$ undergoes the same rotation, and hence $F_g(g\cdot x) = F(x)$.
>
> Thanks for the comment, which raises a good theoretical question. Unfortunately, if $x$ is rotated by matrix $Q$, $L$ does not undergo the same rotation. Here is an example with 6 points: [ 0.687 -0.245  0.713], [ 1.164  1.221 -1.242], [-0.126 -0.797 -0.368], [-0.666 -0.501  1.189], [-0.315 -0.524  0.178], [-0.743  0.848 -0.47 ]
>
> Cholesky Decomposition on the covariance matrix produces $L$, a lower triangular matrix:
>
> [ 0.766  0     0   ]
>
> [ 0.318  0.764  0   ]
>
> [-0.367 -0.496  0.623]
>
> Rotation matrix $Q$:
>
> [-0.059 -0.357 -0.932]
>
> [0.519  0.786 -0.334]
>
> [0.852 -0.503  0.138]
>
> After rotation $Q$, Cholesky Decomposition produces $L_g$:
>
> [0.638  0     0   ]
>
> [0.639  0.904  0   ]
>
> [-0.087  0.033  0.632]
>
> However, $QL \neq L_g$ as shown below!
>
> [0.183  0.19  -0.581]
>
> [0.771  0.767 -0.208]
>
> [0.441 -0.454  0.086]
>
> $QL$ is unnecessarily lower triangular, and thus not the solution of Cholesky decomposition. Specifically, multiple solutions exist for $Q\Sigma Q^ = L_gL_g^T$(including $QL$), but only one unique solution exists (usually not $QL$) if restricted to lower triangular matrices. And we prefer a lower triangular matrix since its inverse matrix is easy to obtain. Hence, we can only derive $L_gL_g^T = QLL^TQ^T$, but not $QL = L_g$.
> > W5: I am surprised that this transformation improves performance. The score model in the transformed space now has to learn that bonds in different orientations should have different lengths, inferring from the inter-residue distances of the binding site how much each dimension is compressed or stretched.
>
> Thanks for your insightful comments! The reasons are two-fold. First, The affine-based normalization enhances generalization, outweighing potential challenges from dimension twisting (please see response to W3). Second, The latent diffusion framework inherently addresses some challenges related to dimension twisting. During diffusion, we generate only latent point clouds with coarse-grained geometries. The detailed full-atom reconstruction is handled by the autoencoder, which operates in the original data space and is encouraged to capture the patterns of bond lengths and angles.
> > Q1: Why do you need to generate all-atom designs? Do you think it could work just as well to generate C-alpha coordinates and discrete residue types?
>
> Thanks for the question! We think atom-level modeling provides better generalization as interactions are determined by secondary bonding between atoms. Ablations in Table 4 show significant degradation in diversity and success rate without full-atom context.
> > Q2: Line 223: does ‘filter out’ mean ‘keep’ or ‘discard’?
>
> Sorry for the confusion. It means 'keep'. We will correct it in the revision.
> > Q3: How could the dataset be augmented?
>
> Thanks for the question! We have explored data augmentation from monomer fragments, showing some enhancement. A synthetic dataset from models like AlphaFold 3 could be possible, but quality and balance issues need to be considered, which we leave for future work.
> > Q4: Table 3: what counts as a good RMSD, for practical purposes? Are all the methods given the same inputs, or are some doing blind docking?
>
> Thanks for the question! Only Alphafold 2 is doing blind docking since it is a cofolding model. For practical purposes, RMSD < 5A ̊  indicates near-native conformations, and 2A ̊ typically indicates high-quality conformations [c].
>
> [c] "Comprehensive evaluation of fourteen docking programs on protein–peptide complexes."Journal of chemical theory and computation 16.6 (2020).
> > Q5: Equation (3): how is the MSE on full-atom structures defined when the residue type is wrong?
>
> During training, we still compare the geometry to the ground truth. Both sequence loss and structure loss guide the model to adjust sequence and structure predictions simultaneously.

---

> > ### Comment · Reviewer_XM8c · 2024-08-12
> > **Thank you for the clarifications**
> >
> > Thank you for the clarifications.
> >
> > Regarding proposition 3.1: you're absolutely right. Sorry, and thank you for your patience.
> >
> > One further detail I could not find in the paper is the exact definition of the binding site. Is this a potential source of data leakage? I am wondering if the binding site is defined in terms of the ground-truth pose of the same peptide.
> >
> > I keep my score as-is.

---

> > > ### Author Response · Authors · 2024-08-12
> > > **Thanks for the thoughtful reviews**
> > >
> > > Thanks sincerely for the thoughtful reviews! For the binding site, we use all residues on the target protein within 10Å to any atoms on the peptide. The threshold 10Å is kind of large to avoid data leakage. We will make this point clear in the revision. Thanks again for your effort in reviewing our submission.

---

### Official Review · Reviewer_xBxR · 2024-07-12

**Soundness:** 3
**Presentation:** 3
**Contribution:** 3
**Rating:** 6
**Confidence:** 4

**Summary:**

This paper explores the structure-based peptide design problem. A benchmark on this task and  a powerful diffusion-based model for full-atom peptide design, named PepGLAD, are proposed. PepGLAD explores the geometric latent diffusion, where the sequence and the full-atom structure are jointly encoded by a variational autoencoder. The proposed method outperforms the existing models on sequence-structure co-design and binding conformation generation.

**Strengths:**

1. This paper tackles an under-explored problem and constructs a new benchmark which is beneficial for this field.
2. The novel Geometric Latent Space can maintain the advantages of Latent Diffusion for efficiency and normalization and Geometric Diffusion for explicit interaction modeling.
3. PepGLAD can achieve full-atom generation compared with backbone atom generation in previous methods, which is truly an improvement.
4. PepGLAD utilizes a novel Receptor-Specific Affine Transformation to map the pockect-specific distribution to the standard distribution

**Weaknesses:**

1. This paper failed to further explore the explicit interaction modeling in geometric latent space. I think it’s the key advantage compared to standard latent space (without the geometric latent).
2. The introduction to Receptor-Specific Affine Transformation can be more clear. I kindly suggest the authors can add one sentence or two summarizing the first paragraph in section 3.3 into the section 1 of Introduction. In such way readers may be more clear that this Affine Transformation is applied to the peptides and the pockets (i.e. binding site distribution in the introduction). Also the point 3 in line 62-65 has a mistake: according to the method section the affine transformation F is mapping binding site distribution (of peptides) into standard Gaussian (not the reverse)
3.  The conditioning method of pockets is not explored. The denoiser takes the geometric latents of peptides and the coordinates of pockets as input, but the latents are coordinates are different levels of abstraction and this may cause some underlying issues.

**Questions:**

1. What’s the motivation for using Latent Diffusion instead of standard diffusion? Just because it’s better in image generation domain?
2. How to incorporate geometric pocket condition (e.g. coordinates) into geometric latent diffusion process?
3. Is the geometric latent are just the Ca atoms of each residues? Will using all four backbone atoms as latent improve the perforemance？
4. Is the adaptive multi-channel equivariant encoder in dyMEAN a scalarization-based equivariant GNNs? It’s better to add some explainations in this paper.
5. Does this Receptor-Specific Affine Transformation beat simple gaussian normalization, common trick in Structure-based Drug Design field?

**Limitations:**

The authors didn’t discuss about the limitations of this method. This paper failed to further explore the explicit interaction modeling in geometric latent space. I think it’s the key advantage compared to standard latent space (without the geometric latent).

---

> ### Author Rebuttal · Authors · 2024-08-06
>
> Thanks for your insightful and constructive comments!
>
> > W1: This paper failed to further explore the explicit interaction modeling in geometric latent space.
>
> We apologize for not making this point clear sufficiently.
>
> Our model inputs the pocket as a condition for diffusion. During denoising, we use an equivariant-GNN-based encoder to capture explicit geometric interactions between the peptide and the pocket. This is achieved by considering distances and inner products of relative positions between the full-atom pockets and the peptide latent points during geometric message passing.
>
> > W2: The introduction to Receptor-Specific Affine Transformation can be more clear. I kindly suggest the authors can add one sentence or two summarizing the first paragraph in section 3.3 into the section 1 of Introduction. In such way readers may be more clear that this Affine Transformation is applied to the peptides and the pockets (i.e. binding site distribution in the introduction). Also the point 3 in line 62-65 has a mistake: according to the method section the affine transformation F is mapping binding site distribution (of peptides) into standard Gaussian (not the reverse)
>
> Thanks for the valuable suggestion!
>
> The 3rd contribution in section 1 tries to summarize the receptor-specific affine transformation. To improve clarity, we will add the following sentence to the paragraph: "While the complexes possess disparate distributions, we derive a receptor-specific affine transformation that is applied to both the binding sites and the peptides, projecting the shape of all complexes into approximately a standard Gaussian distribution."
>
> We also appreciate you pointing out the mistake in lines 62-65. We will correct this in the revision.
>
> > W3: The conditioning method of pockets is not explored. The denoiser takes the geometric latents of peptides and the coordinates of pockets as input, but the latents are coordinates are different levels of abstraction and this may cause some underlying issues.
>
> Thanks for the insightful observation! Since we have applied regularization on the equivariant latent coordinates by using KL divergence with respect to the $C_\alpha$ coordinates in Eq.(4), which helps ensure consistent scales between the peptide latent coordinates and the pocket, mitigating potential issues arising from their different levels of abstraction.  We will highlight this point in the revision to address the reviewer's concern.
>
> > Q1: What’s the motivation for using Latent Diffusion instead of standard diffusion? Just because it’s better in image generation domain?
>
> Thanks for the question!
>
> The motivation for using Latent Diffusion in our work is to effectively address the challenges associated with full-atom generation, which is not well-suited to standard diffusion approaches due to two key problems:
>
> 1. Fixed Data Lengths: In full-atom generation, changing the type of residue during the denoising process would alter the number of atoms per residue. This discrete change in data length is incompatible with standard diffusion framework.
>
> 2. Generation with Paddings: If the number of atoms is fixed to the maximum number, as done in the baseline DiffAb, it is analogous to generating "padding" in NLP, which is suboptimal. Our results in Table 3 show that DiffAb’s performance in recovering full-atom geometry is significantly worse compared to PepGLAD.
>
> In contrast, latent diffusion tackles these issues by using a full-atom encoder to map each residue to a single point in the latent space. This approach avoids changes in data length during diffusion, without need to generate paddings, therefore handles the full-atom generation more elegantly.
>
> > Q2: How to incorporate geometric pocket condition (e.g. coordinates) into geometric latent diffusion process?
>
> Sorry for any confusion. We do incorporate the geometric pocket condition $\mathcal{G}_b$ on both the autoencoder (please see Eq.(1)) and the diffusion model (please see Eq.(11-13)). The pocket is put into the same geometry graph with the peptides for geometric message passing.
>
> > Q3: Is the geometric latent are just the Ca atoms?
>
> We apologize for the ambiguity in the presentation.
>
> The geometric latents are not limited to just the Cα atoms, but regularized around the Cα atoms to prevent exploding coordinates. However, they still encode full-atom geometries, working jointly with invariant latent representations to reconstruct the full-atom geometry.
>
> > Q4: Is the encoder in dyMEAN a scalarization-based equivariant GNNs?
>
> Yes, it is a scalarization-based equivariant GNN since the processing of coordinates in dyMEAN only includes invariant scalars via relative distances or inner products. We will add the explanations in the paper.
>
> > Q5: Does this Receptor-Specific Affine Transformation beat simple gaussian normalization?
>
> Thanks! Yes, our experiments indicate that Receptor-Specific Affine Transformation outperforms simple Gaussian normalization.
> In our preliminary experiments, we found that simple Gaussian normalization does not enhance generalization, as the high variability of peptide coordinate distributions and scales often leads to exploding coordinates on test complexes during generation. As Figure 4 shows, DiffAb, which employs simple Gaussian normalization, exhibits significant variations in RMSD across different test complexes, including many outliers with very large RMSD values. This suggests inferior generalization capability. In contrast, PepGLAD, which utilizes receptor-specific affine transformation, achieves stable RMSD values across various complexes, demonstrating superior generalization ability. Such discrepancy is unique to peptides, which often face more disparate geometry scales compared to antibodies and small molecules.
>
> > The authors didn’t discuss about the limitations of this method
>
> Sorry, we put the discussion on limitations in Appendix J due to limited space. We will move the section to the main text in the revision.

---

> > ### Comment · Reviewer_xBxR · 2024-08-12
> > **Thanks for your rebuttal!**
> >
> > I have read all of the author rebuttal, which addressed most of my concerns. Thus I will raise my score. Thanks for the detailed response.

---

> > > ### Author Response · Authors · 2024-08-12
> > > **Thanks for the insightful reviews**
> > >
> > > Thanks for the insightful reviews, which help refine our manuscript! We will add the responses to the revision to reflect the discussion process. Thanks again for your valuable efforts in reviewing our submission!

---

### Official Review · Reviewer_KHMu · 2024-07-13

**Soundness:** 3
**Presentation:** 3
**Contribution:** 2
**Rating:** 7
**Confidence:** 4

**Summary:**

This paper introduces a novel latent diffusion model, Peptide design with Geometric Latent Diffusion (PepGLAD), for the task of peptide design. The authors propose an affine transformation to project the raw Euclidean space into a standardized one, ensuring physical symmetry. Overall, I think it is a good paper, with solid experimental results and novelty in methods.

**Strengths:**

This paper proposes a latent diffusion model, Peptite design with Geometric LAtent Diffusion (PepGLAD) , applied to the peptide design task. This paper also proposes an an affine transformation to project the raw Euclidean space into a standardized one, so that the physics symmetry can be ensured. The main contributions of PepGLAD over GeoLDM seem to be its application to full-atom peptide generation and the introduction of a receptor-specific affine transformation. While GeoLDM¹¹ was designed for 3D molecule generation, PepGLAD extends this to the more complex scenario of full-atom peptide generation. The receptor-specific affine transformation is a significant contribution, as it projects the geometry to approximately N(0,I) derived from the binding site, which appears to be a novel technique in the field of latent diffusion models for molecules and proteins.

**Weaknesses:**

There are several doubts on method contributions, datasets, as shown in Questions.

**Questions:**

1. What are the key differentiating factors of PepGLAD compared to GeoLDM [1]? While I recognize the shift from latent molecular generation to full-atom peptide generation, are there any specific model design elements (apart from the affine transformation) that set it apart from GeoLDM? Furthermore, does the model incorporate any strategies to reduce the costs associated with generating full-atom peptide scenarios?

2. The most significant contribution made by the authors in this paper is the Receptor-Specific Affine Transformation. This transformation projects the geometry to approximately N(0,I), derived from the binding site. This approach appears to be the first of its kind applied to latent diffusion models for molecules and proteins. I am curious if there are any other works that have used this technique to ensure equivariance in their generation process, such as frame-averaging [6] [7].

3. Compared to previously established databases [2] [3], the datasets utilized in your study appear relatively modest in size. Could you comment on the unique advantages of your proposed datasets over these larger ones? Furthermore, have you considered applying your model to existing datasets to reinforce the validity of your results?

4. I suggest to add more citations on recently proposed related works on peptide design [3] [4] [5].

[1] M Xu, et al. Geometric Latent Diffusion Models for 3D Molecule Generation

[2] Z Wen, et al. Pepbdb: a comprehensive structural database of biological peptide-protein interactions.

[3] L Lin, et al. PPFlow: Target-Aware Peptide Design with Torsional Flow Matching

[4] Osama Abdin, et al. PepFlow: direct conformational sampling from peptide energy landscapes through hypernetwork-conditioned diffusion

[5] Colin A Grambow, et al. RINGER: Conformer Ensemble Generation of Macrocyclic Peptides with Sequence-Conditioned Internal Coordinate Diffusion

[6] W Jing, et al. DSMBind: SE(3) denoising score matching for unsupervised binding energy prediction and nanobody design

[7] Omri Puny, et al. Frame Averaging for Invariant and Equivariant Network Design

**Limitations:**

See Appendix. J Limitations

---

> ### Author Rebuttal · Authors · 2024-08-06
>
> Thanks for your constructive comments!
>
> > Q1: What are the key differentiating factors of PepGLAD compared to GeoLDM [1]? While I recognize the shift from latent molecular generation to full-atom peptide generation, are there any specific model design elements (apart from the affine transformation) that set it apart from GeoLDM? Furthermore, does the model incorporate any strategies to reduce the costs associated with generating full-atom peptide scenarios?
>
> Thanks for the question!
>
> The motivations of our PepGLAD and GeoLDM are fundamentally different. We resort to the latent space where the full-atom geometry of each residue is compressed into a single node so that the diffusion can be implemented. Otherwise, when sampling residues of different type, the number of atoms varies, which is incompatible with the diffusion framework. On the contrary, GeoLDM maps each atom in the small molecule to one point in the latent space, without addressing the problem we are trying to tackle here.
>
> By representing each residue's full-atom geometry as a single point in the latent space, PepGLAD inherently reduces the computational cost associated with generating full-atom scenarios. Specifically, the diffusion process only needs to handle a latent graph which is approximately 10 times smaller than the full-atom graph, resulting in improved efficiency and scalability.
>
> > Q2: The most significant contribution made by the authors in this paper is the Receptor-Specific Affine Transformation. This transformation projects the geometry to approximately N(0,I), derived from the binding site. This approach appears to be the first of its kind applied to latent diffusion models for molecules and proteins. I am curious if there are any other works that have used this technique to ensure equivariance in their generation process, such as frame-averaging [6] [7].
>
> Yes, the proposed technique of receptor-specific affine transformation is novel and the first of its kind to the best of our knowledge. The technique differs from frame averaging from two perspectives. First, the motivations are totally different.  While frame averaging aims to design equivariant models, our technique serves as a normalization skill particularly for pocket-based geometric diffusion models. Second, while frame averaging uses PCA to obtain the principal components of the coordinates, we use cholesky decomposition of the covariance of the coordinates to identify unique invertible affine transformations for projections between the data distribution and standard Gaussian.
>
> > Q3: Compared to previously established databases [2] [3], the datasets utilized in your study appear relatively modest in size. Could you comment on the unique advantages of your proposed datasets over these larger ones? Furthermore, have you considered applying your model to existing datasets to reinforce the validity of your results?
>
> Our dataset is carefully curated to establish two advantages:
>
> 1. Practical relevance: Consistent with previous research [a], we have focused on peptides ranging from 4 to 25 residues in length. This range is particularly relevant to practical applications such as drug discovery, as peptides within this length exhibit favorable biochemical properties [b]. In contrast, existing datasets like PepBDB [c] direct extract all complexes from PDB and lack filtering based on peptide lengths.
>
> 2. Redundancy and split: We remove the redundant complexes and split the dataset according to the sequence identity of the receptor for testing generalization across different target. On the contrary, the existing dataset contains redundant entries and does not provides such splits.
>
> Furthermore, we did apply our model to an existing dataset, PepBDB, to reinforce the validity of our conclusions, the results on which are shown in Table 1 and Table 3.
>
> [a] Tsaban, Tomer, et al. "Harnessing protein folding neural networks for peptide–protein docking." Nature communications 13.1 (2022): 176.
>
> [b] Muttenthaler, Markus, et al. "Trends in peptide drug discovery." Nature reviews Drug discovery 20.4 (2021): 309-325.
>
> [c] Wen, Zeyu, et al. "PepBDB: a comprehensive structural database of biological peptide–protein interactions." Bioinformatics 35.1 (2019): 175-177.
>
> > Q4: I suggest to add more citations on recently proposed related works on peptide design [3] [4] [5].
>
> Thanks for the suggestion! We will definitely discuss and cite these recent works in the revision.

---

> > ### Comment · Reviewer_KHMu · 2024-08-12
> > **Reply**
> >
> > I am glad that my concerns are all address by the authors. And I will lift my rating.

---

> > > ### Author Response · Authors · 2024-08-12
> > > **Thanks for your valuable reviews**
> > >
> > > We sincerely thank the reviewer for the efforts in providing the valuable comments and questions, which helps enhance our submission! We are glad to hear that your concerns have been successfully addressed. We will add the responses to the revision to improve the quality of our paper.

---

### Official Review · Reviewer_8deM · 2024-07-21

**Soundness:** 2
**Presentation:** 3
**Contribution:** 2
**Rating:** 5
**Confidence:** 4

**Summary:**

The authors proposed PepGLAD, a latent diffusion model for full-atom peptide design. This work mainly addressed two challenges for peptide design, (1) full-atom modeling, and (2) diverse binding geometry. Specifically, a variational auto-encoder was trained to learn latent representations of protein-peptide interactions. Then, a latent diffusion model was employed to capture the underlying distribution to sample binding site-conditioned peptide conformations. The authors curated a benchmark dataset to evaluate the model performance, and compared with existing models to showcase its effectiveness.

**Strengths:**

- Full-atom modeling is significant to better represent protein-protein/peptide/ligand interactions. This work goes beyond backbone-only modeling and realized full-atom level generation.
- Employing latent diffusion with a pre-trained VAE is a reasonable choice, which has recently become prevalent for protein generative modeling.
- Provided a new benchmark dataset with careful curation. Proposed novel evaluation metrics and discussed limitations of existing metrics (e.g., AAR).
- Under the proposed evaluation metrics, PepGLAD outperforms existing models.

**Weaknesses:**

- Dataset is small (6105 non-redundant complexes). Data augmentation attempts (ProtFrag) do not seem to help (Table 4 ablation results). Although sequence-based clustering has been utilized to prevent data leakage, training/testing VAE and diffusion model on 6105/93 data points make the results less convincing, *especially considering the enormous design space for sequence-structure co-design tasks*.
- Guidance with an additional classifier (Sec 3.5) makes the model more complicated (which seems a bit ad hoc to me).
- From the results shown in Table1, it seems that the model is able to generate a variety of conformations (high Div.) which are reasonable ($\Delta G < 0$). Given the extremely limited size of dataset, I am not sure if the model can properly capture the *conformation space* at all. In other words, I am a bit skeptical about the proposed evaluation metrics.

**Questions:**

- L#130. For each residue, the initial coordinates are initialized with the same $\vec{\boldsymbol{z}}_i$. How significant are these initial vectors? Can we use an SE(3)-invariant latent space and a regression model to generate atomic coordinates (similar to AlphaFold StructureModule, conditioned on the sampled residue type)?
- Classifier guidance seems ad hoc and brings more complication to modeling. Would an architecture similar to that used in [ESM3](https://www.biorxiv.org/content/10.1101/2024.07.01.600583v1) or [PVQD](https://www.biorxiv.org/content/10.1101/2023.11.18.567666v1) help?
- Table 1: out of 40 samples generated by PepGLAD, there are about 20 different conformations (Div.) and 20 valid peptides (Success). How should the readers interpret this? Peptides exhibit high structural flexibility upon binding?
- Did I miss sequence recovery metrics in the model benchmark section (excluding Consistency, which only models the joint distribution)?

---

> ### Author Rebuttal · Authors · 2024-08-06
>
> Thanks for your valuable efforts and comments!
>
> > W1. Dataset is small. Data augmentation attempts do not seem to help. Although sequence-based clustering has been utilized to prevent data leakage, training/testing VAE and diffusion model on 6105/93 data points make the results less convincing.
>
> Thanks for your comments! Indeed, to ensure our results are convincing, we follow the literature [a] and employ the 93 complexes for testing, which are carefully curated by [a] to represent a diverse and high-quality landscape of complexes. Our model trained on a non-redundant dataset of 6K complexes is able to deliver promising results on these testing complexes. Regarding data augmentation that is only applied to the autoencoder, it has a notable impact on energy scores, without which the success rate drops from 55.97% to 52.15% in Table 4.
>
> In addition, we also conducted experiments on a public dataset PepBDB [b], which is of larger size (10K samples and 190 testing complexes). These results, presented in Table 1 and Table 3, reinforce the validity and robustness of our conclusions.
>
> [a] "Harnessing protein folding neural networks for peptide–protein docking." Nature communications 2022.
>
> [b] "PepBDB: a comprehensive structural database of biological peptide–protein interactions." Bioinformatics 35.1 (2019)
>
> > W2. Guidance with an additional classifier makes the model more complicated.
>
> Thanks! The classifier guidance is used to sample from a sequential subspace within unordered point clouds.  We understand that adding guidance somehow makes our model more complicated, but this component does further enhance the performance. To show this, we have additionally compared the structure prediction model with and without guidance in the table below. As its effect is relatively minor, we did not emphasize it as a major contribution to the paper. We will highlight this point in the revision.
> |guidance|$\text{RMSD}_{C\alpha}\downarrow$|$\text{RMSD}_\text{atom}\downarrow$|$\text{DockQ}\uparrow$|
> |-|-|-|-|
> |w/|4.09|5.30|0.592|
> |w/o|4.10|5.34|0.582|
> > W3. It seems that the model is able to generate a variety of conformations (high Div.) which are reasonable. Given the extremely limited size of dataset, I am not sure if the model can properly capture the conformation space at all. In other words, I am a bit skeptical about the proposed evaluation metrics.
>
> We understand your concern. It is true that using the Diversity metric alone is hard for correct evaluation. Hence, we also use rosetta energy to evaluate the fidelity to check whether the diversity is hacked by random conformations. As highlighted in Table 1, our model surpass the baselines both on diversity and fidelity (rosetta energy), therefore the improvement in diversity should be meaningful as the model is properly capturing the desired distribution. These findings are further confirmed on a larger dataset, PepBDB. For more discussion on the metrics, we kindly refer to appendix G.3.
>
> > Q1: L#130. For each residue, the initial coordinates are initialized with the same. How significant are these initial vectors? Can we use an SE(3)-invariant latent space and a regression model to generate atomic coordinates?
>
> Thanks! The initial vectors from the latent equivariant space are significant, which can convey sufficient geometric informaion. As suggested, we further conduct an experiment by discarding the equivariant latent vectors and using an SE(3)-invariant latent space instead. The table below shows that the reconstruction ability (RMSD, DockQ) of the VAE  deteriorates significantly. It is reasonable since SE(3)-invariant latent space lacks geometric interactions with the pocket atoms, leading to difficulties in reconstructing the full-atom structures on the binding site.
> |Equivariant vectors|AAR$\uparrow$|RMSD$\downarrow$|DockQ$\uparrow$|
> |-|-|-|-|
> |w/|95.1%|0.79|0.898|
> |w/o|93.4%|1.75|0.823|
> > Q2: Classifier guidance seems ad hoc and brings more complication to modeling. Would an architecture similar to that used in ESM3 or PVQD help?
>
> Thanks for the question! Please see our response to W2 for the explanations of the classifier guidance.
> The suggested architectures like ESM3 or PVQD are SE(3)-invariant, which are defect in integrating the geometric conditions of binding sites. They are primarily designed to generate entire proteins with a random coordinate system. However, our pocket-based generation requires the generated structures (i.e. the output) to be located in the same coordinate system as the pocket (i.e. the input), thereby necessitating SE(3)-equivariant modeling.
>
> > Q3: Table 1: out of 40 samples, there are about 20 different conformations (Div.) and 20 valid peptides (Success). How should the readers interpret this? Peptides exhibit high structural flexibility upon binding?
>
> Thanks! To better interpret the results in Table 1, here we additionally calculate the diversity of generated peptides with successful binding (dG<0) in the table below. It reads that our generated peptides exhibit high structural flexibility upon successful binding. Thank you again for raising this valuable question and we will add this new result to the revised paper.
> |Div.|Div. ($\Delta G < 0$)|
> |-|-|
> |0.506|0.632|
> > Q4: Did I miss sequence recovery metrics in the model benchmark section?
>
> Sorry for the confusion. We did analyze the sequence recovery metric, namely, Amino Acid Recovery (AAR) in Appendix G.1 on a large dataset consisting of 600k peptide sequences against 328 receptors. Further, as shown in the table below, a high AAR might not be meaningful or reliable due to the extensive diversity in peptide binding. For example, HSRN achieves an AAR close to our method, but its success rate is much lower. Please find more details in our analyses in Appendix G.1, and we are willing to include the AAR results into the main body if requested.
> |Model|AAR|Success|
> |-|-|-|
> |HSRN|35.8%|10.46%|
> |DiffAb|37.1%|49.87%|
> |PepGLAD (ours)|36.7%|55.97%|

---

> > ### Comment · Reviewer_8deM · 2024-08-12
> > **Response to Rebuttal**
> >
> > Dear Authors,
> >
> > Thanks for your response. Most of my questions have been properly addressed. Although some limitations still remain, e.g., dataset size and benchmark metrics, I think researchers can refer to this work for some insights and know-how to solve their domain specific challenges. To that end, I have raised my rating and support the acceptance of this manuscript.
> >
> > Best,

---

> > > ### Author Response · Authors · 2024-08-12
> > > **Thanks for the constructive reviews**
> > >
> > > Thanks for the constructive comments and questions! We will add the responses, as well as the remaining limitations to the revision, to better enhance the quality of our submission. Thanks again for your efforts in reviewing our submission!

---

### Decision · Program_Chairs · 2024-09-25

**Decision:**

Accept (poster)

**Comment:**

This paper focuses on the problem of designing peptides conditioned on a protein pocket. To that end, the authors first gather a dataset based on the Protein Data Bank, and then propose a diffusion-based model called PepGLAD, which combines several careful modelling and training choices appropriate for the domain.

Reviewers generally considered PepGLAD to be a significant advancement on top of prior work, especially given its ability to do full-atom generation (instead of backbone only). The paper introduces several novel ideas, such as the affine transformation between the pocket position and a normal distribution, which most reviewers found interesting and appropriate for the downstream task. Apart from the model itself, this work also carefully curates a dataset with a reasonable train/test split; the reviewers noted the dataset is quite small, but they still considered it to be an important contribution to the field. While the initial version of the paper sparked many clarification questions, these were mostly addressed by the authors during the rebuttal period. In particular, one of the reviewers noted a subtle point around how the protein pocket is defined during data generation (which could potentially "leak" the shape of the ground-truth peptide); the authors did consider this nuance, and their current approach seems satisfactory.

In summary, this paper curates a useful dataset for an important research problem, and builds an effective model on that dataset, combining a mix of novel and pre-existing ideas. I recommend the paper to be accepted, but I would invite the authors to take the clarification questions from the reviewers seriously, in order to make sure the manuscript is more accessible. In particular, for the final version of the paper I would encourage the authors to include a description of how the protein pocket is defined and what potential issues could arise from not doing this carefully.